# Dissecting mechanisms of ligand binding and conformational changes in the glutamine-binding protein

Zhongying Han[1], Sabrina Panhans[1], Sophie Brameyer[2], Ecenaz Bilgen[3], Marija Ram[1], Anna Herr[1], Alessandra Narducci[1], Michael Isselstein[1], Paul David Harris[4], Oliver Brix[1], Pazit Con[1,5], Kirsten Jung[2], Don C Lamb[3], Eitan Lerner[4,6], Douglas Griffith[1], Thomas R Weikl[7]*, Niels Zijlstra[1]*, Thorben Cordes[1,8]*

[1]Physical and Synthetic Biology, Faculty of Biology, Ludwig-Maximilians-Universität München, Munich, Germany; [2]Microbiology, Faculty of Biology, Ludwig-Maximilians-Universität München, Munich, Germany; [3]Department of Chemistry, Ludwig-Maximilians-Universität München, Munich, Germany; [4]Department of Biological Chemistry, The Alexander Silberman Institute of Life Sciences, Faculty of Mathematics & Science, The Edmond J. Safra Campus, The Hebrew University of Jerusalem, Jerusalem, Israel; [5]Department of Poultry and Aquaculture, Institute of Animal Sciences, Agricultural Research Organization, Volcani Center, Rishon LeZion, Israel; [6]The Center for Nanoscience and Nanotechnology, The Hebrew University of Jerusalem, Jerusalem, Israel; [7]Department of Biomolecular Systems, Max Planck Institute of Colloids and Interfaces, Potsdam, Germany; [8]Biophysical Chemistry, Department of Chemistry and Chemical Biology, Technische Universität Dortmund, Dortmund, Germany

*For correspondence:
thomas.weikl@mpikg.mpg.de (TRW);
n.zijlstra@gmail.com (NZ);
thorben.cordes@tu-dortmund.de (TC)

## eLife Assessment

This **important** study combines a comprehensive range of biophysical, kinetic, and thermodynamic techniques, together with high-quality experimental and computational analysis, to carry out a series of well-designed experiments to explore whether glutamine-binding protein binds glutamine via an induced fit or a conformational selection process. The evidence supporting the major conclusion of the work is **compelling**. The work will be of broad interest to biochemists and biophysicists.

**Abstract** The glutamine-binding protein GlnBP is part of an ATP-binding cassette transporter system in *Escherichia coli* and uses two well-characterized conformational states, an open ligand-free and a closed-liganded state, to facilitate active amino-acid uptake. Existing literature on its ligand-binding mechanism lacked sufficient evidence to univocally assign the kinetic type of binding mechanism for GlnBP: ligand binding prior to conformational change, that is an induced fit, or the conformational selection, in which the ligand binds the matching conformation from a pre-existing ensemble. Since such mechanistic questions are relevant for our fundamental understanding of how this and other biomacromolecules regulate cellular processes, we here revisit the question for GlnBP. We present a biochemical and biophysical analysis using a combination of calorimetry, single-molecule and surface-plasmon resonance spectroscopy, and molecular dynamics simulations. We found that both apo- and holo-GlnBP show no detectable exchange between open and (semi-)closed conformations on timescales between 100 ns and 10 ms and that ligand binding and conformational changes in GlnBP are correlated. A global analysis of our experimental results suggests

that the conformational selection model is only compatible with GlnBP for the extreme scenario of very fast conformational exchange between the open and closed states on timescales <100 ns. In contrast, all data remains compatible with an induced-fit mechanism, where the ligand binds GlnBP prior to conformational rearrangements. Importantly, our work demonstrates that it is an intricate task to identify the type of kinetic binding mechanism and that this requires not only a sufficient set of data, but also an integrative experimental and theoretical framework to address the question. Based on this concept, we propose that various protein systems, for which so far only insufficient kinetic data are available, should be revisited.

## Introduction

Periplasmic substrate-binding proteins (SBPs; *Quiocho and Ledvina, 1996*; *Shilton et al., 1996*; *Hall et al., 1997*; *Skrynnikov et al., 2000*; *Trakhanov et al., 2005*; *Wang et al., 2012*) are small, soluble proteins (molecular weight <100 kDa) that are often associated with membrane complexes, including the superfamily of ATP-binding cassette (ABC) transporters (*Mächtel et al., 2019*). SBPs recognize and bind numerous classes of substrates including (but not limited to) ions, vitamins, co-factors, sugars, peptides, amino acids, system effectors, and virulence factors (*Scheepers et al., 2016*). Major biological functions of SBPs are to facilitate membrane transport by delivery of substrate molecules to a transmembrane component or to signal the presence of a ligand (*Scheepers et al., 2016*; *Berntsson et al., 2010*). They are ubiquitous in archaea, prokaryotes, and eukaryotes and possess a highly conserved three-dimensional architecture with two rigid domains, D1/D2, that are linked by a flexible hinge composed of β-sheets (*Figure 1A*), α-helices, or smaller sub-domains (*Scheepers et al., 2016*; *Berntsson et al., 2010*). The available crystal structures of SBPs reveal that many exist in a minimum of two distinct conformations, a ligand-free open (apo) and a ligand-bound closed state (holo; *Figure 1B*).

Several recent studies focused on the characterization of structural dynamics and conformational heterogeneity as well as the ligand-binding mechanisms that underlie SBP function (*Van Meervelt et al., 2017*; *Kim et al., 2013*; *Seo et al., 2014*; *Gouridis et al., 2015*; *Husada et al., 2015*; *Feng et al., 2016*; *Zhang et al., 2020*; *de Boer et al., 2019b*; *Ploetz et al., 2021*). Based on crystal structures, it was proposed that SBPs use a venus-flytrap ligand-binding mechanism in which binding traps the ligand via its closed conformation (*Scheepers et al., 2016*; *Berntsson et al., 2010*). Provided that both ligand binding and conformational changes are well separated in their timescales, the venus-flytrap corresponds to an induced-fit (IF) ligand-binding mechanism (*Figure 1C*). In IF, the event of ligand binding triggers the functionally relevant conformational change. This intuitive model was challenged by nuclear magnetic resonance (NMR)-based paramagnetic relaxation enhancement (PRE) experiments (*Tang et al., 2007*), molecular dynamics (MD) simulations (*Bucher et al., 2011a*; *Bucher et al., 2011b*), and X-ray crystallography (*Flocco and Mowbray, 1994*; *Oswald et al., 2008*; *Ortega et al., 2012*), revealing the existence of unliganded closed- or semi-closed states and their dynamic exchange with the respective open (apo) conformation. Similar findings on ligand-independent

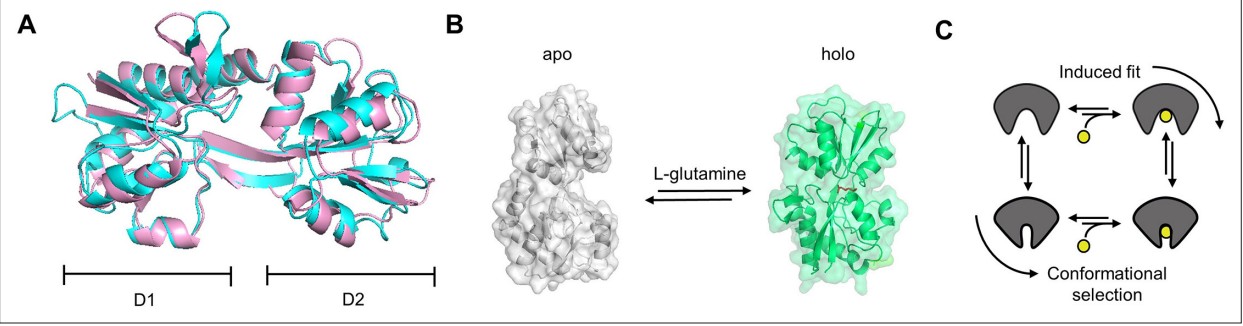

**Figure 1.** Conformational states and possible ligand-binding mechanisms of typical SBPs. (**A**) Structural comparison of SBD2 from *Lactococcus lactis* (PDB file:4KR5 *Fulyani et al., 2013*; cyan) and glutamine-binding protein GlnBP from *E. coli* (pink). SBD2 and GlnBP share 34% sequence identity with a TM-score of 0.90, indicating high structural similarity. (**B**) Crystal structures of the ligand-free (PDB file:1GGG *Hsiao et al., 1996*; grey) and ligand-bound (PDB file:1WDN *Sun et al., 1998*; green) state of GlnBP from *E. coli*. (**C**) Sketch of ligand binding via induced-fit (IF) and conformational selection (CS).

conformational changes were presented for the maltose-binding protein, MalE (*Bucher et al., 2011a*; *Bucher et al., 2011b*; *Stockner et al., 2005*), histidine-binding protein (HisJ; *Jayanthi et al., 2020*), D-glucose/D-galactose-binding protein (GGBP; *Flocco and Mowbray, 1994*; *Ortega et al., 2012*; *Luck and Falke, 1991*; *Careaga et al., 1995*), ferric-binding protein (FBP; *Atilgan and Atilgan, 2009*), choline/acetylcholine SBP (ChoX; *Oswald et al., 2008*), the Lysine-, Arginine-, Ornithine-binding (LAO) protein (*Oh et al., 1994*), and glutamine-binding protein (GlnBP; *Feng et al., 2016*; *Sun et al., 2005*; *Pang et al., 2003*). For MalE, NMR techniques revealed a low-abundance (<10%) semi-closed state that is in rapid dynamic equilibrium with the open apo conformation on the <50 μs timescale. The existence of closed, unliganded state(s) permits an alternative mechanism via conformational selection (CS; *Paul and Weikl, 2016*), where conformational changes occur intrinsically and prior to ligand binding. In CS, the ligand selects the relevant state, for example closed, for binding (*Figure 1C*). IF and CS represent the simplest kinetic schemes to describe the coupling of conformational changes and ligand (un)binding against which available kinetic data can be tested to falsify the types of ligand-binding mechanisms.

Ligand-binding mechanisms have also been the focus of several studies of GlnBP (*Feng et al., 2016*; *Zhang et al., 2020*; *Wu et al., 2022*). GlnBP is part of an ABC transporter system in *E. coli* and binds L-glutamine with sub-micromolar affinity (*Pistolesi and Tjandra, 2012*; *Su et al., 2007*) and arginine with millimolar affinity (*Chen et al., 2020*). It is monomeric and comprised of two globular domains: the large domain (residues 5–84, 186–224) and the small domain (residues 90–180), linked via a flexible hinge (residues 85–89, 181, and 189). GlnBP was crystallized in two distinct conformational states: open (apo, ligand-free; *Hsiao et al., 1996*) and closed (holo, ligand-bound; *Sun et al., 1998*; *Pistolesi and Tjandra, 2012*; *Figure 1B*). GlnBP (*Feng et al., 2016*; *Zhang et al., 2020*; *Wu et al., 2022*) has recently been studied by a combination of single-molecule Förster resonance energy transfer (smFRET), NMR residual dipolar coupling (RDC; *Feng et al., 2016*) experiments, MD simulations (*Sun et al., 2005*; *Pang et al., 2003*), and Markov state models (MSMs; *Wu et al., 2022*). Based on the results, it was proposed that GlnBP undergoes pronounced conformational changes both in the absence (*Feng et al., 2016*) and in the presence (*Zhang et al., 2020*) of substrate, involving a total of four to six conformational states. These findings lead to the interpretation that ligand binding in GlnBP could occur by means of a combination of CS and IF (*Wu et al., 2022*), which we found controversial in light of other existing data demonstrating that NMR experiments on GlnBP do not support the idea of intrinsic conformational dynamics in apo-GlnBP (*Kooshapur et al., 2019*). Furthermore, the observation of multiple GlnBP conformers under apo- and holo-conditions via smFRET (*Feng et al., 2016*; *Zhang et al., 2020*; *Wu et al., 2022*) does not align with findings from MD simulations (*Kienlein and Zacharias, 2020*) and smFRET work of substrate binding domains SBD1 and SBD2 (from the amino acid transporter GlnPQ *de Boer et al., 2019b*; *Gouridis et al., 2021*), which structurally resemble GlnBP (*Figure 1A*; SBD2 shows ~34% sequence identity with GlnBP, TM-score of 0.90). Finally, ligand binding to the fully closed state of the GlnBP conformation seems rather unlikely, considering the limited accessibility of the binding site, which is also seen for related proteins such as MalE (*Telmer and Shilton, 2003*).

Such controversial findings and arguments reveal a central problem in the study of ligand-binding mechanisms, which is the availability of sufficient experimental evidence to distinguish one mechanism from the other. Importantly, both IF and CS imply temporal order of ligand-protein interactions and conformational changes and thus require kinetic data for univocal identification (*Paul and Weikl, 2016*; *Weikl and Paul, 2014*; *Vogt and Di Cera, 2012*; *Daniels et al., 2014*; *Gianni et al., 2014*; *Chakrabarti et al., 2016*). The existence of a ligand-free protein conformation, which structurally resembles a ligand-bound form, is necessary (*Ortega et al., 2012*; *Bouvignies et al., 2011*; *Mondal et al., 2018*) but by itself not sufficient evidence for a CS mechanism, as ligand binding may not proceed via this conformation at all. Vice versa, the inability to experimentally detect ligand-free closed conformations cannot be taken as an indicator for IF as a dominant pathway (*Paul and Weikl, 2016*; *Weikl and Paul, 2014*; *Vogt and Di Cera, 2012*; *Daniels et al., 2014*; *Gianni et al., 2014*; *Chakrabarti et al., 2016*) since only very few techniques are able to detect low abundance (high free energy) conformers and their exchange kinetics with the stable ones. Whether, for example, a ligand-free closed (or near-closed) conformation (*Oldham et al., 2007*) can be observed depends on the magnitude of its equilibrium probability (*de Boer et al., 2019b*) as well as the sensitivity of the techniques used to probe it. While single-molecule fluorescence approaches can provide such

information (*Kim et al., 2013*; *Seo et al., 2014*; *Gouridis et al., 2015*; *Husada et al., 2015*; *Feng et al., 2016*; *Zhang et al., 2020*; *de Boer et al., 2019b*), they often suffer from photon-limited time-resolution and potential labeling artifacts. Analysis of ensemble-averaged relaxation rates from nonequilibrium stopped-flow kinetics or equilibrium NMR experiments can provide the appropriate time resolution, but may be inconclusive under certain experimental conditions (*Chakrabarti et al., 2016*; *Chakrabarti et al., 2022*), for example under the pseudo-first-order condition of high ligand concentrations in stopped-flow experiments (*Paul and Weikl, 2016*; *Weikl and Paul, 2014*; *Vogt and Di Cera, 2012*; *Daniels et al., 2014*; *Gianni et al., 2014*; *Chakrabarti et al., 2016*). Consequently, to validate or rule-out the presence of a certain ligand-binding mechanism such as IF and CS, a set of complementary and consistent structural, thermodynamic, and kinetic data of the protein system is required in combination with theoretical model building (*Chakrabarti et al., 2016*).

Here, we revisit the question of the ligand-binding mechanisms for GlnBP by biochemical and biophysical analyses of ligand binding and its coupling to conformational changes. For this, we used a combination of isothermal titration calorimetry (ITC), smFRET, surface-plasmon resonance (SPR) spectroscopy, and MD simulations to derive sufficient evidence that can support the (in)compatibility of the data with any of two kinetic mechanisms. Using smFRET, we observed that apo- and holo-GlnBP show no detectable exchange with other conformational states on timescales between 10 ms and 100 ns. Any observed FRET dynamics could be traced back to photophysical origins rather than to conformational changes. Importantly, in all our smFRET assays, ligand binding and conformational dynamics are highly correlated. In MD simulations of GlnBP, we observed ligand unbinding only after or during the transition from the closed to the open protein conformation. A global analysis of our experimental

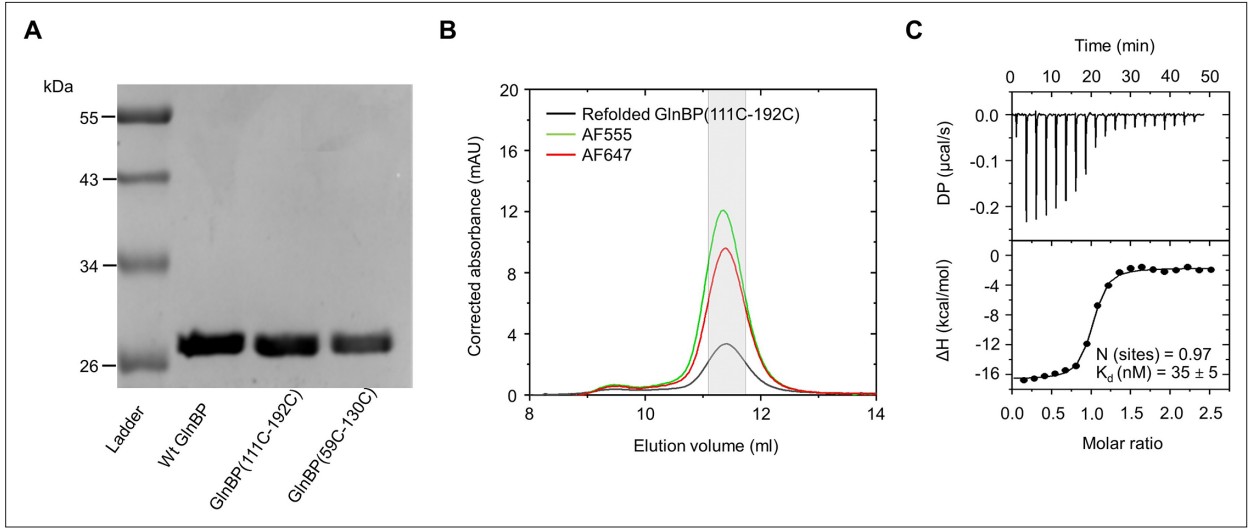

**Figure 2.** Biochemical characterization, fluorescence labeling, and thermodynamic characterization of ligand binding of GlnBP. (**A**) SDS-PAGE analysis of GlnBP purity with Coomassie-staining. Lane 1, molecular mass ladder with sizes of proteins indicated in kDa; lane 2, purified GlnBP WT; lane 3, purified double-cysteine variant GlnBP(111 C-192C); lane 4, purified double-cysteine variant GlnBP(59 C-130C). (**B**) SEC was used to further purify the fluorescently-labeled proteins. The protein absorption was monitored at 280 nm (black curve), the donor dye absorption (AF555) at 555 nm, and the acceptor dye absorption (AF647) at 647 nm. The labeling efficiency of AF555 and AF647 was estimated to be 71% and 59%, respectively. For the solution-based smFRET measurements, the used protein fractions are indicated in grey. (**C**) Ligand-binding affinities of refolded, unlabeled GlnBP(111 C-192C) were determined by ITC with a $K_d$ = 35 ± 5 nM for L-glutamine (mean value from N=3 with standard deviation), which is in agreement with previous reports (*Chen et al., 2020*). The free energy of binding was ΔG=−42.6 kcal/mol with the enthalpy ΔH=−62.3 kcal/mol and entropy contributions - T*ΔS=19.9 kcal/mol.

The online version of this article includes the following source data and figure supplement(s) for figure 2:

**Source data 1.** PDF file containing SDS page gel for *Figure 2A*, indicating the relevant bands and treatments.

**Source data 2.** File containing the original gel for *Figure 2A*.

**Figure supplement 1.** Crystal structure and dye accessible volume calculations of GlnBP cysteine variants.

**Figure supplement 2.** Size-exclusion chromatography (SEC) of refolded GlnBP WT and GlnBP variants.

**Figure supplement 3.** Investigating binding affinity of refolded GlnBP WT and refolded GlnBP(59 C-130C) using isothermal titration calorimetry (ITC) measurements.

results suggests that the CS model is only compatible with GlnBP for the extreme scenario of very fast conformational exchange between the open and closed states on timescales <100 ns. In contrast, all data remains compatible with an IF mechanism, where the ligand binds GlnBP prior to conformational rearrangements.

## Results

### Biochemical characterization of GlnBP and ligand binding

For our study, we produced wild-type protein GlnBP (GlnBP WT) and two double-cysteine variants for analysis of conformational states via smFRET: GlnBP(111 C-192C) with point mutations at V111C and G192C (*Figure 2—figure supplement 1*) and GlnBP(59 C-130C) with point mutations at T59C and T130C (the latter was adapted from *Feng et al., 2016*; *Zhang et al., 2020*; *Figure 2—figure supplement 1C*). All protein variants were expressed in *E. coli* and purified using affinity chromatography (see Materials and methods for details). Protein purity was assessed by Coomassie-stained SDS-PAGE analysis (*Figure 2A*). As reported previously, GlnBP co-purifies with bound glutamine (*Tjandra et al., 1992*), which was removed by unfolding and refolding of the purified protein. We verified the monomeric state and proper folding of the resulting protein using size-exclusion chromatography (SEC, *Figure 2B*) by comparing the elution volume and shape of the monodisperse peak of GlnBP before and after the procedure (*Figure 2—figure supplement 2*).

To assess the binding affinity of GlnBP WT and the two GlnBP cysteine variants for L-glutamine, we performed ITC (*Velázquez-Campoy et al., 2004*). Refolded GlnBP WT showed a $K_d$ for L-glutamine of 22±7 nM (*Figure 2—figure supplement 3A*) and $K_d$ values of 31±3 nM and 35±5 nM for the two cysteine variants (*Figure 2C*, *Figure 2—figure supplement 3B*). These values are in overall agreement with previously published data (*Chen et al., 2020*). This verifies that the unfolding and refolding process as well as cysteine substitutions did not impact the biochemical properties of unlabeled GlnBP.

### Analysis of conformational states of freely diffusing GlnBP via smFRET

After assessing the thermodynamic properties of GlnBP, we characterized the conformational states and changes associated with ligand binding via smFRET. With smFRET, it is possible to study biomacromolecules in aqueous solution at ambient temperature, and identify conformational changes, heterogeneity, small sub-populations, and determine microscopic rates of conformational changes (*Lerner et al., 2021*; *Lerner et al., 2018*; *Hellenkamp et al., 2018*). We performed smFRET experiments on freely diffusing (*Figure 3*) and surface-immobilized GlnBP using the refolded variants GlnBP(111C-192C) and GlnBP(59C-130C) labeled with two different dye pair combinations, AF555/AF647 and ATTO 532/ATTO 643, to assess any position- and fluorophore-dependent effects. The smFRET assays were designed such that the inter-dye-distance of the apo state results in a lower FRET efficiency as compared to the holo state of the protein (*Figure 2—figure supplement 1A/C*).

Solution-based μsALEX (*Lerner et al., 2018*) data of GlnBP(111 C-192C) labeled with AF555/AF647 are shown in *Figure 3B* after an all-photon burst search (*Nir et al., 2006*). Both apo and holo states, in the absence and presence of saturation levels of glutamine, respectively, show a clear predominant population of donor-acceptor-labeled protein at S*-values of ~0.5, with two distinct mean apparent E* values for the apo (mid FRET, 0.51) and holo (high FRET, 0.68) states (*Figure 3B/C*, *Figure 3—figure supplement 1A*). This can be interpreted as a transition from open (apo) to closed (holo) GlnBP conformations upon the addition of the ligand. Similar results were obtained for the second double-cysteine variant (GlnBP(59 C-130C), *Figure 3D*, *Figure 3—figure supplement 2*) and from measurements with a different pair of fluorescent dyes for GlnBP(111 C-192C) (ATTO 532/ATTO 643; *Figure 3—figure supplement 1B*). Notably, further analysis and a comparison of mean accurate FRET efficiencies and the inter-dye distances show good agreement with simulated inter-dye distances of ±0.2 nm using the open- and closed-GlnBP crystal structures except for the holo-state of GlnBP(59 C-130C), which deviated by ~0.8 nm (*Figure 2—figure supplement 1B/D*).

Importantly, a quantitative analysis of the fraction of the closed state (high-FRET) subpopulation as a function of ligand concentration (*Figure 3C and D*) with a simple binding isotherm and no cooperative effects (n=1) provides $K_d$ values in the 20–50 nM range for all labeled GlnBP variants. These results are fully consistent with ITC (*Figure 2C*, *Figure 3—figure supplements 1/3*). Interestingly, we found that arginine, the non-cognate ligand of GlnBP, induces hardly any FRET shifts even at

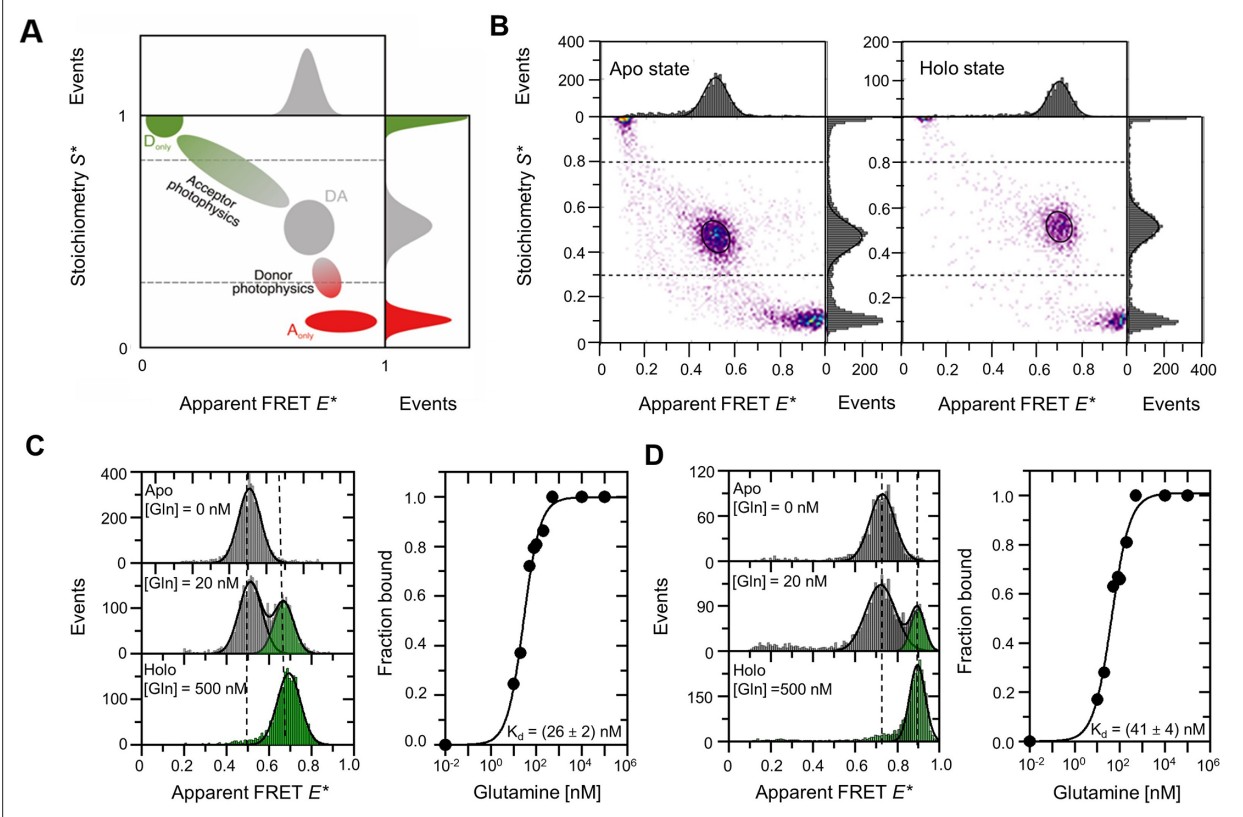

**Figure 3.** smFRET analysis of GlnBP using diffusion-based μsALEX. (**A**) Graphical depiction of an E*-S* histogram obtained by μsALEX; panel reproduced from Figure 3C from *Eiring et al., 2021*. Using μsALEX, the stoichiometry S* can be used to separate donor-only (S>0.8, D_only), acceptor-only (S<0.3, A_only), and the FRET molecular species with both donor and acceptor fluorescently active fluorophore (S* between 0.3–0.8, DA). Bridge artifacts or smearing caused by donor or acceptor photophysics (photoblinking and/or photobleaching) can cause artificial broadening of the FRET population or a shift of the extracted mean apparent FRET efficiency. (**B**) μsALEX-based E*-S* histograms of the refolded GlnBP(111 C-192C) double-cysteine variant labeled with AF555 and AF647. (**C, D**) Diffusion-based single-molecule analysis and ligand-binding affinity measurements with μsALEX of doubly-labeled GlnBP(111 C-192C) (**C**) and GlnBP(59 C-130C) variants (**D**) at different ligand concentrations. Values provided are mean +/- SD (N=3). For plotting purposes, the concentration of glutamine in the apo state was set artificially to a value of 0.01 nM in the right parts of panels C/D. Data fitting of the fraction of the high-FRET subpopulation as a function of ligand concentration was performed with the Hill equation for n=1, which is a valid approximation for describing the bound fraction of GlnBP as a function of glutamine in the case where [GlnBP] <<$K_d$.

The online version of this article includes the following figure supplement(s) for figure 3:

**Figure supplement 1.** L-glutamine-induced conformational changes in refolded GlnBP(111 C-192C) visualized by μsALEX measurements.

**Figure supplement 2.** L-glutamine-induced conformational changes in refolded GlnBP(59 C-130C) visualized by μsALEX measurements.

**Figure supplement 3.** Investigating binding affinities of fluorescently labeled GlnBP variants using smFRET measurements.

**Figure supplement 4.** Conformational states of refolded GlnBP variants probed by solution-based μsALEX measurements reveal nearly unchanged conformations.

**Figure supplement 5.** Investigating L-Arginine binding affinity of refolded GlnBP(111 C-192C) and GlnBP(59 C-130C) variants using isothermal titration calorimetry (ITC) measurements.

millimolar concentrations of ligand (*Figure 3—figure supplement 4*) despite its binding to GlnBP at these concentrations (*Figure 3—figure supplement 5*).

We were also unable to identify a clear high-FRET subpopulation in the absence of a ligand, ruling out intrinsic conformational dynamics on timescales slower than the burst duration, >10 ms (*Figure 3B*, apo). To estimate an upper bound of the fraction of poorly sampled low abundance states, we determined the percentage of bursts outside of the main FRET population in the range of <E*>±σ of the characteristic FRET population of apo GlnBP(111 C-192C). For this, the FRET populations were fitted with Gaussian functions (with mean values and σ), which serves as a good approximation for mean E* values. For representative data sets of AF dyes, we found ~12% bursts outside of the main

peak range (E*$_{holo}$ = 0.64, σ=0.061, N=626; E*$_{apo}$ = 0.47, σ=0.070, N=5,013) and for ATTO dyes, ~4% bursts outside the peak region (E*$_{holo}$ = 0.56, σ=0.056, N=124; E*$_{apo}$ = 0.37, σ=0.047, N=2,908). This suggests an upper bound of 5–10% for a subpopulation of other FRET subpopulations, for example partially closed conformations of GlnBP. Thus, our results agree with the idea that GlnBP mainly exists in one state – the open conformation – in the absence of its ligands.

## Screening for rapid conformational dynamics via analysis of 'within-burst' FRET dynamics

Next, we analyzed our smFRET data for 'within-burst' dynamics using burst-variance analysis (BVA; *Torella et al., 2011*), multi-parameter photon-by-photon hidden Markov modeling (mpH²MM; *Harris et al., 2022*), intensity-based FRET efficiency versus donor lifetime ($E - \tau$; E stands for FRET efficiency, $\tau$ is lifetime) plots (*Kalinin et al., 2010*) and burst-wise fluorescence correlation spectroscopy (FCS). These analyses provide access to FRET-dynamics that occur on timescales from a few milliseconds down to the sub-µs regime. This allows us to assess whether the observed FRET populations represent stable conformational states or time averages of (rapidly) interconverting states.

We first performed BVA of GlnBP(119-192) data with ATTO 532/ATTO 643 as a dye pair using a dual-channel burst search (DCBS; *Nir et al., 2006*). In BVA, within-burst E*-dynamics are identified as an elevated SD of the apparent FRET efficiencies, σ(E*), beyond what is expected from photon statistics, that is σ(E*) values larger than the theoretical semicircle (*Figure 4—figure supplements 1–3*, panels A). Our analysis indicates that, for each of the ligand concentrations, at least some of the recorded single molecules undergo dynamic changes in E* while diffusing through the confocal spot (*Figure 4—figure supplement 1*). The within-burst dynamics are more prominent for dyes AF555 and AF647 (*Figure 4—figure supplement 2*) and become most abundant in the variant GlnBP(59C-130C), which was used in previous studies (*Feng et al., 2016*; *Zhang et al., 2020*; *Wu et al., 2022*; *Figure 4—figure supplement 3*). It is important to note that dynamic changes in apparent FRET efficiency can have photophysical origins and do not necessarily confirm the presence of conformational dynamics. For example, the apparent dynamic changes in E* might represent within-burst dynamics between FRET-active subpopulations (i.e. S*~0.5) and FRET-inactive subpopulations (e.g. donor-only, acceptor-only). Therefore, it is essential to quantify the BVA observed dynamics and identify the corresponding E*-S* subpopulations between which the dynamic transitions occur.

For this purpose, we used multi-parameter photon-by-photon hidden Markov modeling (mpH²MM; *Harris et al., 2022*; *Pirchi et al., 2016*) to identify the most-likely state model that describes the experimental results based on how E* and S* values change within single-molecule bursts. Such analysis can provide rates of exchange between distinct states of E*/S* and its interpretation is described in detail in Appendix 1. The mpH²MM analyses can differentiate whether apparent dynamic changes in E* arise from two conformational subpopulations or from photophysical transitions that do not represent conformational dynamics of GlnBP. Our analysis in *Figure 4B* shows clear signatures for donor- and acceptor-blinking between bright and dark states of the fluorophores (*Figure 4*, *Figure 4—figure supplements 1–3*), that is the FRET species with intermediate S* exchange with species of very high and low S* values, respectively. mpH²mm identifies a single and static apo FRET-active mid-E* state in the absence of a ligand and a single and static FRET-active high-E* state in the presence of saturating levels of ligand, which we ascribe to the open (mid-E*) and closed (high-E*) conformations of GlnBP. It is only in the presence of low concentrations of glutamine (around its K$_d$) where two FRET-active subpopulations, representing two distinct conformational states, are identified that might interconvert on timescales slower than 10ms (i.e. slower than typical burst durations). In conclusion, if intrinsic conformational dynamics existed in apo or holo GlnBP, it could only be between the highly populated FRET conformation we identify and another conformation that is populated significantly below the sensitivity of our measurement and analysis (i.e. a minor population with a fraction <5–10%) or these transitions would have to occur much faster than the time resolution of our experiments (<100 µs), which is dictated by the alternation periods in the µsALEX experiment.

To check for the presence of faster dynamics, we used multiparameter fluorescence detection with pulsed interleaved excitation (MFD-PIE) (*Kudryavtsev et al., 2012*). GlnBP(119-192) labeled with ATTO 532/ATTO 643 were used since this combination showed the least photophysical artifacts. In *Figure 5*, we first show two-dimensional plots of FRET efficiency (E) versus donor fluorescence lifetime values in the presence of acceptor ($\tau_{D(A)}$) for apo and holo GlnBP. The theoretical linear relationship

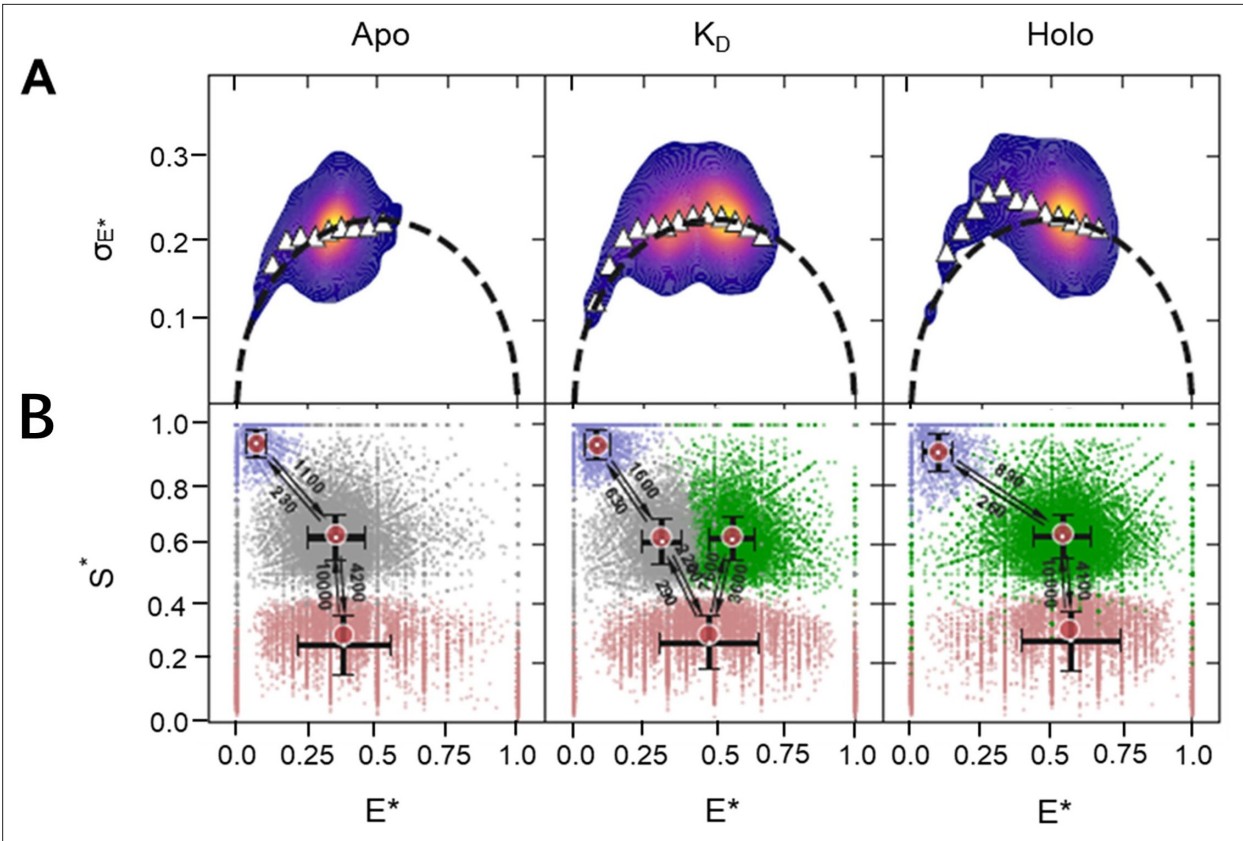

**Figure 4.** Screening GlnBP for rapid within-burst FRET dynamics. (**A**) Burst-variance analysis (BVA) showing a weak signature of within-burst FRET dynamics in the low E* regime. (**B**) Two-dimensional E* versus S* scatter plots of dwells in mpH$^2$MM-detected states within bursts detected by the Viterbi algorithm. Arrows and adjacent numbers indicate transition rates in s$^{-1}$. Transitions with rates <100 s$^{-1}$ are omitted since such long dwells in a state before transitions are improbable to occur within single-molecule bursts with durations <10 ms and are most probably a mathematical outcome of the mpH$^2$MM optimization framework. The dispersion of the E* and S* values of dwells in mpH$^2$MM-detected states are due to the short dwell times in these states, where the shorter the dwell time in a state is, the lower the number of photons it will include, and hence the larger the uncertainty will be in the calculation of E* and S* values of dwells. E* and S* are E* and S* values uncorrected for background, since in mpH$^2$MM all burst photons are considered, including ones that might be due to background. Full analysis shown in *Figure 4—figure supplement 1*.

The online version of this article includes the following figure supplement(s) for figure 4:

**Figure supplement 1.** Screening GlnBP(111 C-192C) for rapid within-burst FRET dynamics.

**Figure supplement 2.** Screening GlnBP(111 C-192C) for rapid within-burst FRET dynamics.

**Figure supplement 3.** Screening GlnBP(59 C-130C) for rapid within-burst FRET dynamics.

between E and $\tau_{D(A)}$ defines the static FRET line (*Figure 5A*, black lines). When the labeled molecules exhibit dynamics faster than the diffusion time, the fluorescence-weighted-average of the donor lifetime becomes biased towards longer donor lifetimes due to the higher brightness values of low-FRET species (*Kalinin et al., 2010*). Therefore, fast conformational switching is seen as bursts with distinct FRET efficiency values exhibiting a population shift towards the right of the static FRET line. As can be observed from the $E - \tau$ plots (*Figure 5A*), the center of mass of the FRET populations for both apo and holo GlnBP are coinciding with the static FRET line, suggesting the absence of conformational changes on timescales faster than milliseconds in line with data in *Figure 4*.

We also looked for dynamics using burst-wise FCS analysis (*Figure 5B*). For this, bursts containing signal from both fluorophores were selected, padded with 50 ms before and after burst identification, and the fluorescence autocorrelation functions of donor (*Figure 5B*, green curves) and acceptor signals (*Figure 5B*, red curves), as well as for the fluorescence cross-correlation functions between donor and acceptor signals (*Figure 5B*, black curves), were calculated. Conformational dynamics are expected to manifest themselves as an anticorrelation contribution in the cross-correlation function between donor and acceptor channels due to fluctuations in FRET efficiencies that occur faster than

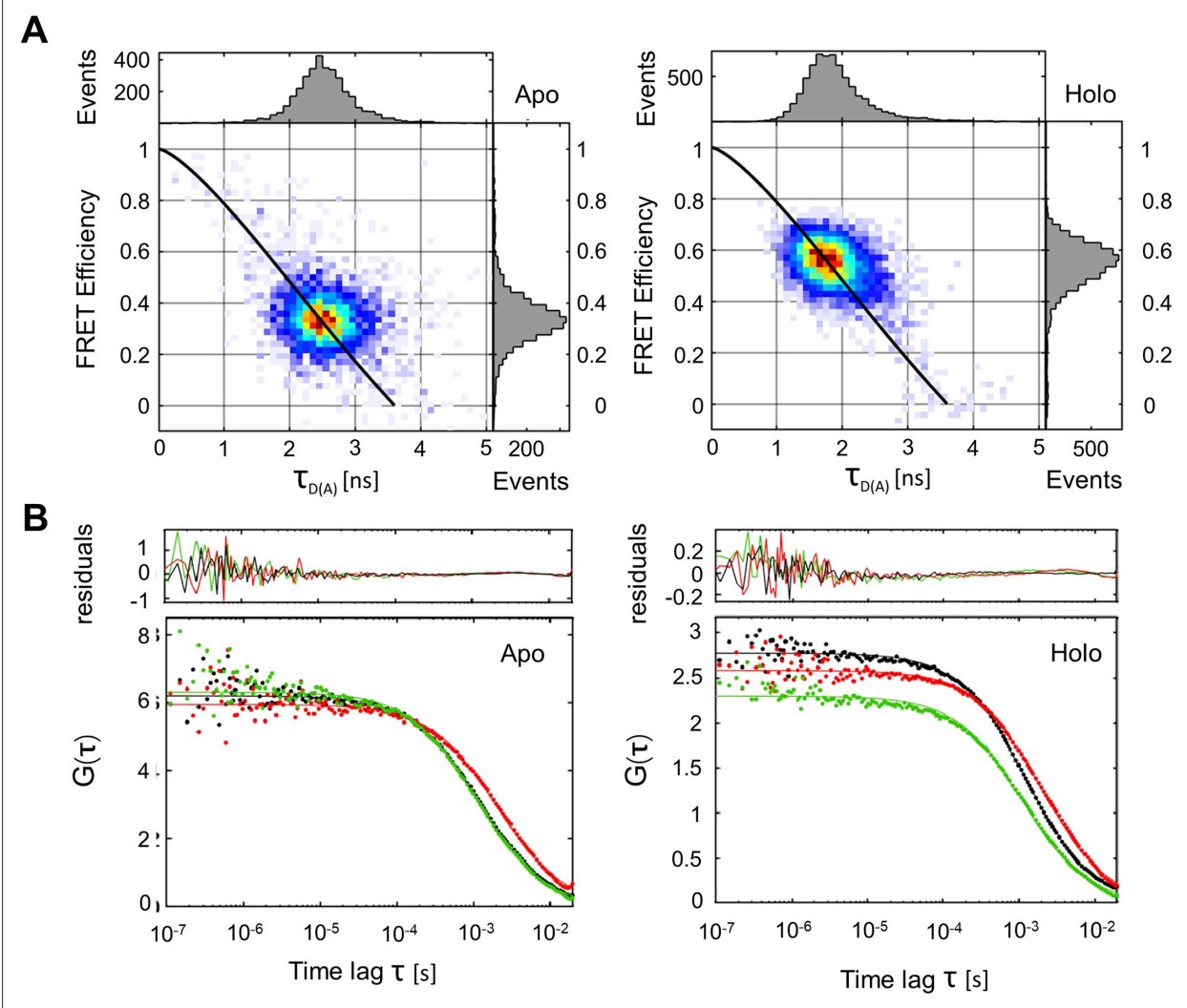

**Figure 5.** Screening GlnBP for rapid dynamics within single molecule bursts using $E - \tau$ and burst-wise FCS analyses. (**A**) Two-dimensional histogram of FRET efficiency (E) versus donor lifetime in the presence of acceptor ($\tau_{D(A)}$) for apo (left) and holo (right) GlnBP. The FRET populations coincide well with the theoretical static FRET line (black) indicating the absence of conformational dynamics taking place at timescales faster than ms. (**B**) Analysis of FRET conformational dynamics using burst-wise FCS for apo and holo states on bursts exhibiting photoactive donor and acceptor fluorophores. The fluorescence autocorrelation functions of the detected donor (DDxDD) and acceptor signal (AAxAA) are displayed in green and red, respectively. The fluorescence cross-correlation function between donor and acceptor signals (DDxDA) is shown in black.

the translational diffusion component of the correlation functions (~1 ms on our setup; *Torres and Levitus, 2007*). The burst-wise FCS analysis at times <100 µs resulted in plateaued cross-correlation functions (*Figure 5B*, black lines) for apo and holo states, indicating the lack of dynamics down to the time-resolution of the experiments, that is the typical clock time of the photon time tagging on the order of 100 ns. It has to be mentioned that minor population exchange concerning <10% of molecules cannot be excluded with absolute certainty, particularly for the time regime <10 µs. Here, the noise increases due to limited photon budget, yet no clear indication for a cross-correlation related to conformational changes is seen, also supported by non-systematic fluctuations in the residuals (*Figure 5*).

## Studies of surface-immobilized GlnBP via TIRF microscopy

Next, we characterize GlnBP and its conformational dynamics on timescales beyond the residence time of molecules in the confocal excitation volume (i.e. >1–10 ms) with the hope to obtain information on rare conformational events. We consequently conducted smFRET with NTA-based

surface-immobilization of the GlnBP His-tag using TIRF microscopy (see Appendix 2 and accompanying *Appendix 2—figures 1–5* for details). We reasoned that this would also allow the direct comparison of our results to those of Wang, Yan, and co-workers (*Feng et al., 2016*; *Zhang et al., 2020*; *Wu et al., 2022*). Importantly, in our analysis, we found that various buffer additives used for oxygen depletion have the same effect on GlnBP as the addition of glutamine (i.e. apo-GlnBP becomes artificially 'closed' in the presence of the additives) as we demonstrated in solution-based μsALEX experiments (*Appendix 2—figure 1*). Consequently, these additives were omitted since their effects mimic that of substrate binding. Strikingly, the conformational states of GlnBP were also partially altered upon surface immobilization (*Appendix 2—figure 2*), that is the E* values of GlnBP in apo/holo-state were significantly higher than in solution (*Appendix 2—figures 2 and 4*). Furthermore, GlnBP did not retain its full biochemical activity on the glass coverslips (*Appendix 2—figure 2*), that is only ~50% of all GlnBP molecules showed the expected shift towards higher FRET values upon addition of the ligand (*Appendix 2—figure 2F*). To validate our setup and immobilization approach, we additionally tested dsDNA (*Appendix 2—figure 2A,C*) and the two previously studied proteins SBD1 and SBD2 (*Appendix 2—figure 5*). Here, we did not observe discrepancies in FRET efficiency or biochemical activity, and the data of freely diffusing and surface-immobilized species were consistent.

Our combined smFRET analysis of GlnBP under different biochemical conditions suggests that conformational changes are tightly coupled to the ligand glutamine (*Figure 3*). We can also rule out prominent conformational dynamics on timescales between 100 ns and 10ms of apo and holo GlnBP

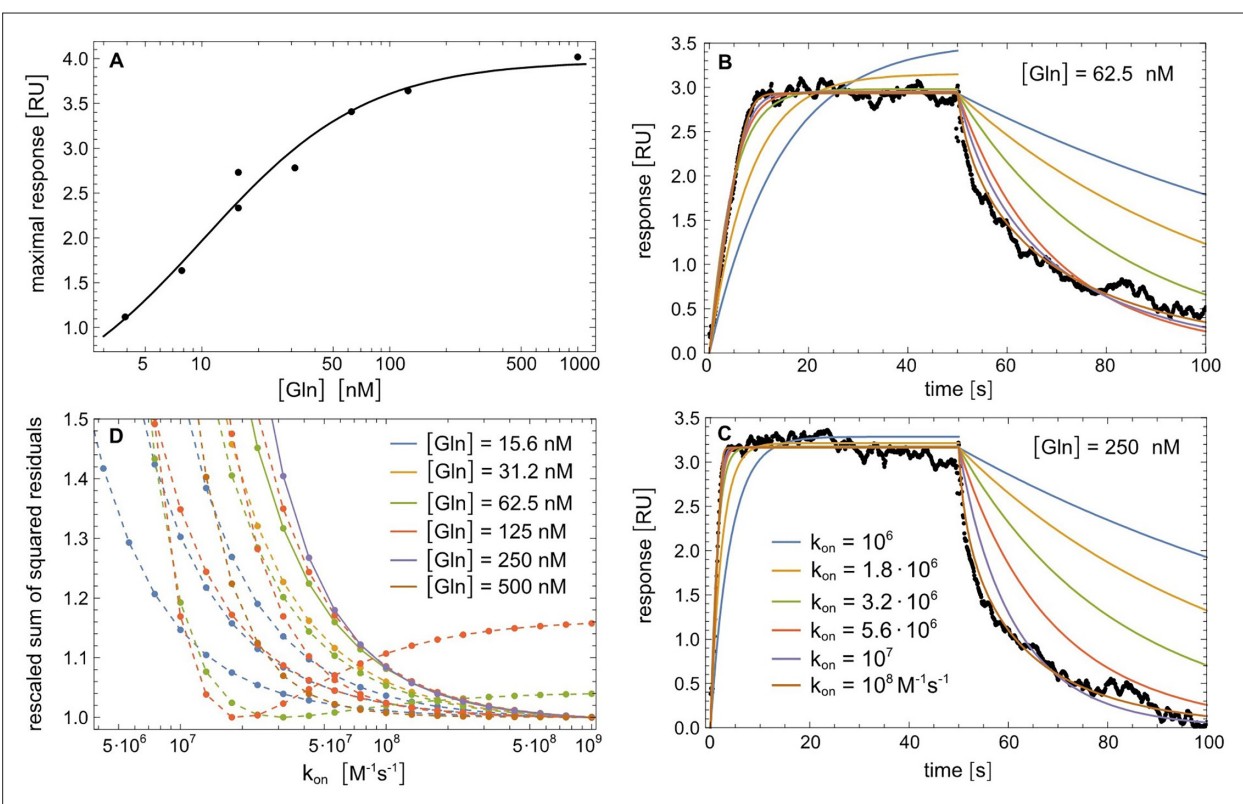

**Figure 6.** Kinetic analysis of ligand binding and dissociation in GlnBP using SPR. (**A**) Fitting of maximal responses in sensorgrams from a measurement set with [Gln] concentrations from 7.8 to 1000 nM (data points) with $f = c/\left(1 + K_d/\left[\text{Gln}\right]\right)$ leads to $K_d = 10 \pm 1$ nM. (**B, C**) SPR sensorgrams with an association phase of 50 s at the indicated glutamine concentrations [Gln] followed by a dissociation phase of 50 s with [Gln]=0 in the bulk flow (data points), and fits of the sensorgrams with the reaction scheme (1) for different values of the effective on-rate constant $k_{on}$ (see Materials and methods for details). (**D**) Rescaled sum of squared residuals versus $k_{on}$ for fits of sensorgrams with different values of [Gln] in the association phase. Note that multiple repeats for the ligand concentrations [Gln]=15.6 nM, 62.5 nM, and 125 nM are plotted. The two curves with full lines correspond to fits in panels B and C. The 11 curves with dashed lines correspond to the fits in *Figure 6—figure supplement 1*. Note that panels are arranged in clockwise order.

The online version of this article includes the following figure supplement(s) for figure 6:

**Figure supplement 1.** SPR sensorgrams at the indicated glutamine concentrations, and fits of the sensorgrams for different values of the effective on-rate constant, $k_{on}$, as in *Figure 6B/C*.

via mpH$^2$MM, MFD-PIE, and burst-wise FCS (*Figures 4 and 5*). Furthermore, our analysis suggests that apo GlnBP does not adopt (partially) closed conformations on the timescale >10 ms with an abundance >5–10% (*Figures 3–5*). While these results provide valuable information on ligand-binding affinity, conformational heterogeneity, and timescales of conformational dynamics in GlnBP, they are insufficient to exclude one or the other kinetic ligand-binding mechanism (IF vs. CS). We thus decided to integrate the obtained information into a general theoretical framework for analysis of ligand-binding mechanisms (*Paul and Weikl, 2016*; *Weikl and Paul, 2014*; *Vogt and Di Cera, 2012*), for which additional knowledge of the association and dissociation rates of ligand binding are required.

## Insights on ligand-binding kinetics from bulk spectroscopy

Such kinetic information is available from SPR spectroscopy or stopped-flow experiments. Since SPR was available to us, we immobilized GlnBP via its His-tag on a sensor chip and monitored its interaction with glutamine as a function of time. Even though GlnBP became partially inactive during immobilization for smFRET in TIRF microscopy (*Appendix 2—figures 1–5*), we reasoned that non-functional GlnBP will not be observed in SPR since only functional protein can contribute to the signal changes. The assumption that GlnBP remains functional on SPR-chips was validated by the match of ligand-binding characteristics obtained from ITC (*Figure 2C*, *Figure 2—figure supplement 3*), smFRET (*Figure 3C and D*), and SPR (*Figure 6A*).

In SPR, GlnBP showed specific and stable interaction with glutamine based on the magnitude of the equilibrium RU response as a function of glutamine concentration (*Figure 6A*). Analysis of the concentration-dependent maximal RU units yields a $K_d$ of 10 nM (*Figure 6A*). The overall maximal response of around 3–4 RU indicates a 1:1 stoichiometry of glutamine assuming a monomeric state of GlnBP (*Figures 2C and 6*). Kinetic association and dissociation experiments were conducted under pseudo-first order conditions, that is the assumption of constant glutamine concentrations during an SPR run, due to the applied flow of buffer. The data were analyzed with the standard two-step reaction scheme (*Schuck and Zhao, 2010*; *Schasfoort, 2017*):

$$\text{Gln}_{\text{bulk}} \underset{k_t}{\overset{k_t}{\rightleftharpoons}} \text{Gln}_{\text{surface}} + \text{GlnBP} \underset{k_{\text{off}}}{\overset{k_{\text{on}}}{\rightleftharpoons}} \text{GlnBP-Gln} \qquad (1)$$

This includes a mass-transport step between the bulk solution of the applied flow and the sensor surface with transport rate $k_t$ in both directions, and a binding step with effective on- and off-rate constants, $k_{\text{on}}$ and $k_{\text{off}}$. Because of the dominance of mass transport, fits of this reaction scheme to the SPR sensorgrams using fit parameters $k_t$ and $k_{\text{on}}$ (after substituting $k_{\text{off}}$ with $K_d \, k_{\text{on}}$ in the scheme) do not allow determination of $k_{\text{on}}$ within reasonable error bounds. However, fits with fixed values of $k_{\text{on}}$ indicate that effective on-rate constants smaller than $10^7$ M$^{-1}$s$^{-1}$ are incompatible with the sensorgrams (*Figure 6B,C*, *Figure 6—figure supplement 1*). More precisely, plots of the rescaled sum of squared residuals for these fits versus $k_{\text{on}}$ (*Figure 6D*) indicate a lower bound of at least $3 \cdot 10^7$ M$^{-1}$s$^{-1}$ for $k_{\text{on}}$; this implies $k_{\text{off}} = K_d \, k_{\text{on}} > 0.3$ s$^{-1}$ (with $K_d = 10$ nM). Among the 13 plots in *Figure 6D*, and among the 4 plots for [Gln]=125 nM, only 1 plot exhibits a minimum of the sum of squared residuals below this bound and is therefore likely an outlier.

## Sequences of events along MD simulation trajectories

To further investigate the coupling between conformational changes of GlnBP and ligand binding/unbinding, we performed atomistic simulations starting from the ligand-bound GlnBP structure with the AMBER20 software implementation for graphics processing units (GPUs; *Case et al., 2020*; *Salomon-Ferrer et al., 2013*) and the ff14SB force field parameters (*Schuck and Zhao, 2010*; *Maier et al., 2015*; see Materials and methods for details). To observe unbinding events on the microsecond timescale accessible in our simulations, we reduced all interactions between the protein and the ligand by 16%. With these reduced interactions, we observed ligand unbinding and a conformational change from the closed to the open GlnBP conformation in 5 out of 20 simulation trajectories with a length of 2 µs. *Figure 7* illustrates characteristic distances for GlnBP opening and ligand unbinding on these 5 trajectories for time windows of 500 ns around the unbinding point. GlnBP opening is monitored by the distances between the C-α atoms of the residues 117 and 137 in domain 2 and the residue 51 in domain 1. We chose these residue pairs because they exhibit large relative changes in distance, with distances of 4.5 and 7.5 Å between residues 51 and 117 and residue 51 and 137 in the closed GlnBP

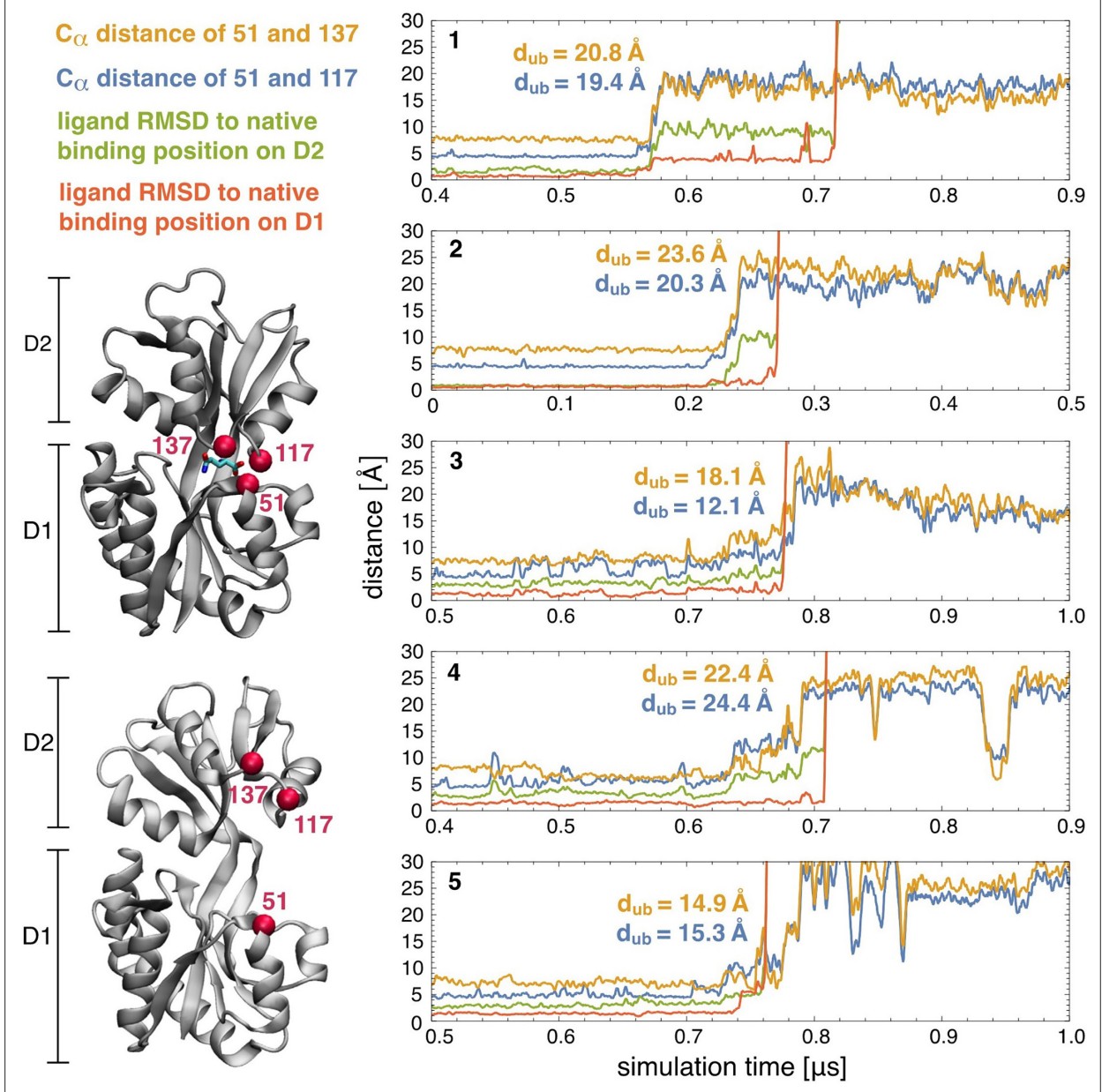

**Figure 7.** Protein conformational changes and ligand unbinding along MD simulation trajectories. Characteristic distances reflecting protein opening (blue, yellow) and ligand unbinding (green, red) within time windows of 500 ns around the unbinding point of 5 out of 20 trajectories with a total length of 2 µs starting from the closed protein-ligand complex. On the 15 other trajectories, the protein remained in the closed ligand-bound state. To observe unbinding on the microsecond timescales accessible in the simulations, the interactions between the protein and ligand were reduced by 16% in the simulation model (see Materials and methods). The distances $d_{ub}$ are the distances between the $C_\alpha$ atoms of the residues 51 and 117 (blue) and 51 and 137 (yellow) at the ligand unbinding point, i.e. at the time point at which the ligand RMSD to the native binding position on domain D1 (red) exceeds 10 Å.

The online version of this article includes the following figure supplement(s) for figure 7:

**Figure supplement 1.** Opening transitions along MD trajectories starting from the closed, ligand-free protein conformation.

conformation, respectively, and distances between about 15 and 30 Å in the open conformation for both pairs. Ligand unbinding is monitored by the root mean square deviation (RMSD) between the non-hydrogen atoms of the ligand in the simulation structures and the ligand in the bound crystal structure, after alignment of either the D1 or the D2 protein domain of these structures. These two RMSDs quantify the distance of the ligand to its native binding position on D1 and D2, respectively.

In trajectories 1, 2, and 4 in *Figure 7*, ligand unbinding occurs clearly after the opening transition of the protein, in agreement with the IF pathway of *Figure 1C*. During the opening transition of these trajectories, the ligand RMSD to the native binding position on D2 increases, which reflects the breaking of the ligand contacts to D2 during opening. The ligand RMSD to the native binding position on D1 remains low until the unbinding point, at which also the ligand contacts to D1 break. On the trajectories 3 and 5, in contrast, the ligand already unbinds during the opening transition of the protein, but also only after substantial opening at distances $d_{ub}$ of the residues 117 and 137 to residue 51 at the unbinding point that are much larger than the corresponding distances in the bound conformation. It is important to note that the reduction of the protein-ligand interactions in our simulations lowers the binding free energies of the two protein conformations rather homogeneously, akin to a reduction of the ligand concentration, and reducing the ligand concentration is known to shift the flux towards the conformational-selection route of *Figure 1C* (*Weikl and von Deuster, 2009*; *Hammes et al., 2009*; *Michel, 2016*), if parallel pathways are possible.

Based on the simulations, IF seems the dominant binding mechanism also for the original, un-rescaled protein-ligand interactions of our atomistic model, because the sequences of opening and unbinding events observed on our simulations at weakened interactions clearly point towards IF, and because the weakening of the interactions rather decreases than increases the tendency for IF in *Figure 1C*. For the original, un-rescaled protein interactions, we expect significantly longer dwell-times in the closed state, significantly longer times for ligand unbinding from D1 after domain unbinding compared to the trajectories 1, 2, and 4 of *Figure 7*, and a significantly lower probability for ligand unbinding already at protein opening as on trajectories 3 and 5.

In addition, we performed simulations starting from the closed GlnBP structure with removed ligand to explore the conformational dwell times in the exchange between the closed and open conformation in the ligand-free state. We observed transitions from the closed to the open GlnBP conformation on 11 out of 20 MD simulation trajectories with a length up to 3 µs (see *Figure 7—figure supplement 1*). On the remaining nine trajectories, the closed conformation persisted for the simulation length of 3 µs. The fraction P(t) of trajectories that exhibit an opening transition up to timepoint t points towards a mean dwell time of several hundred nanoseconds for the closed conformation in the ligand-free state (*Rhee et al., 2004*). On the 11 trajectories that exhibited opening transitions, no subsequent transitions back to closed conformation were observed, which indicates clearly longer dwell-times in the open ligand-free GlnBP conformation of the simulations.

## Discussion

Conformational states of macromolecular complexes and changes thereof govern numerous cellular processes including replication (*van Oijen and Loparo, 2010*), transcription (*Ebright and Busby, 1995*; *Chakraborty et al., 2012*), translation (*Blanchard, 2009*), signal transduction (*Cox and Der, 2010*; *Smock and Gierasch, 2009*; *Tzeng and Kalodimos, 2009*), membrane transport (*Locher, 2016*; *Noinaj and Buchanan, 2014*), regulation of enzymatic activity (*Kamerlin and Warshel, 2010*; *Petrović et al., 2018*; *Loveridge et al., 2012*; *Huse and Kuriyan, 2002*), and the mode of action of molecular motors (*Vale, 2003*; *Cabezon et al., 2012*). While many conformational changes that are triggered by ligand binding have been characterized extensively, it has also become evident that proteins exhibit prominent intrinsic structural dynamics without the involvement of ligands or other biomacromolecules (*Quiocho and Ledvina, 1996*; *Shilton et al., 1996*; *Hall et al., 1997*; *Skrynnikov et al., 2000*; *Trakhanov et al., 2005*; *Wang et al., 2012*; *Henzler-Wildman et al., 2007*; *Vishwakarma et al., 2018*; *Fernandes et al., 2021*; *Maslov et al., 2023*; *Gabba et al., 2014*; *Calabrese et al., 2020*). Elucidating the kinetic binding mechanisms of proteins and biomolecules will advance our understanding of their fundamental mechanisms and allow the identification of critical steps that might allow rational design of selective and effective inhibitors.

In a four-state system (*Figure 1C*), ligand-binding can occur via two 'extreme' kinetic pathways, that is ligand binding occurs before conformational change (IF) or conformational change occurs before ligand binding (CS). The clear temporal ordering of ligand binding and conformational change along either of these pathways implies that the binding transition times, that is the times the ligand needs to enter and exit the protein binding pocket, are small compared to the dwell times of the protein in the two conformations. An important notion is that the ligand-binding mechanisms IF/CS only require temporal separation of ligand binding and conformational changes and are independent of the type

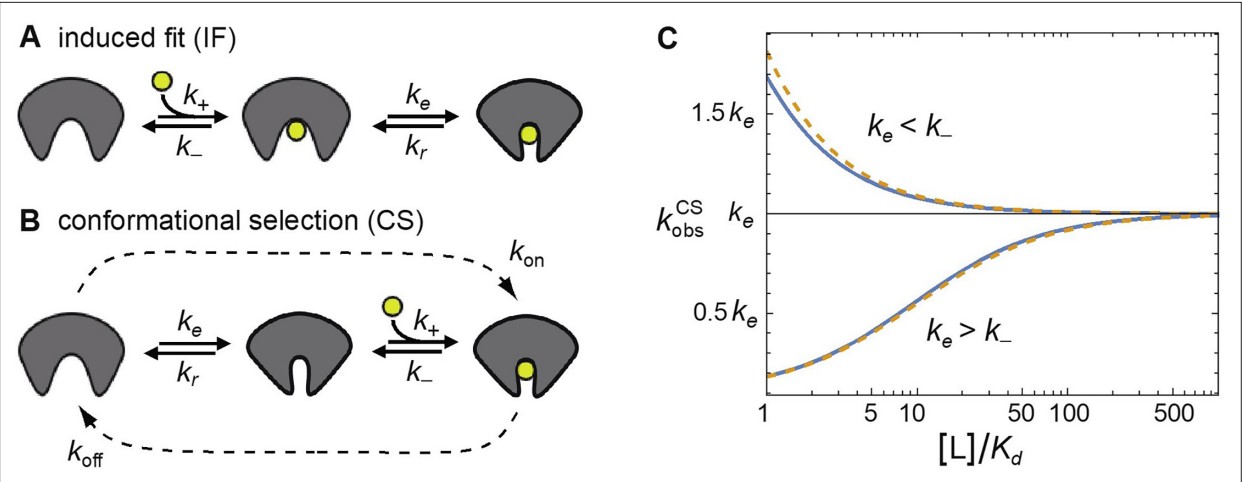

**Figure 8.** Kinetic description of IF and CS pathways and dominant relaxation rate $k_{obs}^{CS}$ of the CS pathways. (**A, B**) Induced-fit and conformational-selection pathways with conformational excitation and relaxation rates, $k_e$ and $k_r$, and with association and dissociation rate constants, $k_+$ and $k_-$, for the binding-competent conformation of the pathway. (**C**) Dominant relaxation rate, $k_{obs}^{CS}$, of the conformational-selection pathways versus ligand concentration [L]. Blue lines represent the exact pseudo-first-order result $k_{obs}^{CS} = \frac{1}{2}\left(S - \sqrt{S^2 - 4\left(k_e\left(k_+\left[L\right] + k_-\right) + k_r k_-\right)}\right)$ with $S = k_e + k_r + k_+\left[L\right] + k_-$ and $K_d = k_-\left(k_e + k_r\right)/k_+ k_e$ for $k_- = 10k_e$ and $k_r = 9\,k_e$ (upper curve) and $k_- = 0.1\,k_e$ and $k_r = 9\,k_e$ (lower curve). The dashed yellow lines represent the approximate result from **Equation 2**. For the induced-fit pathway, the dominant relaxation rate $k_{obs}^{IF} = \frac{1}{2}\left(S - \sqrt{S^2 - 4\left(k_+\left[L\right]\left(k_e + k_r\right) + k_e k_-\right)}\right)$ with S as above is monotonically increasing (similar to $k_{obs}^{CS}$ for $k_e > k_-$) and has the limiting value $k_e + k_r$ at large ligand concentration (**Paul and Weikl, 2016**; **Vogt and Di Cera, 2012**).

The online version of this article includes the following figure supplement(s) for figure 8:

**Figure supplement 1.** Exemplary plots of the dominant relaxation rate $k_{obs}^{IF}$ versus ligand concentration [L] for rate parameters consistent with **Equation 7**.

of conformational motion found in the specific protein. While the concrete conformational motion can be distinct for different SBPs, for example a one- or two-domain motion, spring hammer type of motion (**Chandravanshi et al., 2021**), the type of conformational motions need not be confused with a kinetic ligand-binding mechanism IF/CS.

For GlnBP, we dissected the ligand-binding processes and conformational dynamics using complementary techniques. We used smFRET experiments to monitor dynamics of conformational changes, SPR to monitor ligand binding and dissociation kinetics, we obtained ligand affinity values from ITC, SPR, and smFRET, and explored sequences of conformational opening and ligand unbinding events on simulation trajectories starting from the bound complex. GlnBP fulfills all criteria to use either an IF or CS mechanism since the essential temporal ordering of binding and conformational changes is plausible for SBPs due to their small ligands (**Weikl and Paul, 2014**) and confirmed by the simulation data in **Figure 7**. Since IF/CS represent the simplest kinetic schemes to describe the coupling of conformational changes and ligand (un)binding, we firmly believe that testing available data against these should be the first step before constructing more complex networks of states.

We consequently ask the question, which binding mechanism is compatible with all the data. We hereby follow a published theoretical framework that aims at an unambiguous assignment of the reaction schemes via kinetic rate analysis (**Paul and Weikl, 2016**; **Weikl and Paul, 2014**; **Vogt and Di Cera, 2012**). In essence, we test whether the experimental parameters are compatible both with the IF pathway and the CS pathway (**Figure 1**) or only one of them. Both pathways are shown in **Figure 8A and B** with the relevant kinetic parameters, that is conformational excitation and relaxation rates, $k_e$ and $k_r$, and with association and dissociation rate constants, $k_+$ and $k_-$, for the binding-competent conformation.

Our smFRET analysis indicates that ligand binding is correlated to a conformational change from an open to a closed state of GlnBP and gives detailed information on the conformational dynamics. It excludes prominent structural dynamics of apo- and holo-GlnBP on timescales above 100 ns. While we cannot explicitly rule out conformational exchange of minor sub-populations of potential ligand-free

(partially-)closed conformations and apo-GlnBP, we estimate an upper bound for such processes of <10%. The analysis of SPR sensorgrams leads to the bounds $k_{on} > 3 \cdot 10^7$ M$^{-1}$s$^{-1}$ and $k_{off} = K_d\, k_{on} > 0.3$ s$^{-1}$ (with K$_d$ = 10 nM) for the effective on- and off-rate constants $k_{on}$ and $k_{off}$ at all considered ligand concentrations of glutamine up to 500 nM.

Based on this information, we first discuss the scenario of a dominant CS pathway in GlnBP. To relate it to the effective on- and off-rates of the SRP analysis, we note that the relaxation rate $k_{obs}^{CS}$ of the CS reaction scheme in **Figure 8B** can be well-approximated by:

$$k_{obs}^{CS} \approx k_{on}[L] + k_{off} \tag{2}$$

Here, the effective on- and off-rate constants are $k_{on} = k_e k_+ / \left( k_r + k_+ \left[L\right] \right)$ and $k_{off} = k_r k_- / \left( k_r + k_+ \left[L\right] \right)$ that depend on the conformational transition rates, $k_e$ and $k_r$, between the open and closed conformation in an unbound GlnPB and on the rates, $k_+$ and $k_-$, for the binding step in the closed conformation along this pathway. This approximation holds for small populations of the closed conformation in ligand-free GlnPB with an upper bound of 10% from the smFRET analysis and for ligand concentrations [L]>$K_d$ and, thus, for all the concentrations shown in **Figure 6** (**Weikl and Paul, 2014**). At the largest ligand concentration of 500 nM of the SPR sensorgrams, we obtain $k_{obs}^{CS} > 15$ s$^{-1}$ from this equation, using a lower limit of $3 \cdot 10^7$ M$^{-1}$s$^{-1}$ for $k_{on}$. **Equation 2** can be further simplified to

$$k_{obs}^{CS} \approx \frac{k_e k_- \left( 1 + \left[L\right]/K_d \right)}{k_e + k_- \left[L\right]/K_d} \tag{3}$$

with $K_d = k_- k_r / k_+ k_e$. The limiting value of $k_{obs}^{CS}$ at large ligand concentration [L] obtained from this equation is $k_e$. To conclude the argument, we now consider two cases: (1) for $k_e > k_-$, the relaxation rate $k_{obs}^{CS}$ increases with [L] as seen in **Figure 8C** (lower curve). The limiting value $k_e$ of $k_{obs}^{CS} \left( \left[L\right] \right)$ is therefore larger than 15 s$^{-1}$, because $k_{obs}^{CS} > 15$ s$^{-1}$ at $\left[L\right] = 500$ nM (see above). (2) for $k_e < k_-$, the relaxation rate $k_{obs}^{CS}$ decreases with [L] as seen in **Figure 8C** (upper curve). In this case, $k_{obs}^{CS} \left( \left[L\right] \right)$ is already very close to its limiting value $k_e$ at $\left[L\right] = 500$ nM for $K_d = 10$ nM. In both cases, we thus obtain $k_e > 15$ s$^{-1}$, and from this, $k_r > 9\, k_e > 135$ s$^{-1}$ for an upper bound of 10% of the population $k_e/k_e + k_r$ of the closed conformation in ligand-free GlnBP. However, rates $k_r > 135$ s$^{-1}$ correspond to transition timescales <7.4 ms, which are timescales for conformational dynamics of apo GlnBP that are precluded by the smFRET results presented here. Alternatively, timescales smaller than 100 ns are 'allowed' for the conformational exchange between the open and closed state, which is theoretically possible, but not in line with our MD results of the conformational exchange in the apo state (**Figure 7—figure supplement 1**).

In contrast to the limited validity of the CS mechanism for very fast exchange between the open and closed state, IF is fully compatible with all experimental data presented here. Similar to **Equation 2** for IF, we can approximate the relaxation rate $k_{obs}^{IF}$ in SPR:

$$k_{obs}^{IF} \approx k_{on}[L] + k_{off} \tag{4}$$

Here, the effective on- and off-rate constants are $k_{on} = k_+ k_r / \left( k_- + k_r \right)$ and $k_{off} = k_- k_e / \left( k_- + k_r \right)$. From the equation for the effective off-rate constant $k_{off}$, we obtain

$$k_- = k_{off} k_r / \left( k_e - k_{off} \right) \tag{5}$$

which implies

$$k_{off} < k_e \tag{6}$$

From our SPR results in **Figure 6**, we concluded a lower bound of $3 \cdot 10^7$ M$^{-1}$s$^{-1}$ for $k_{on}$ in a range of ligand concentrations [L] from 15.6 to 500 nM, which likely holds also for smaller [L]. Based on stopped-flow mixing experiments of GlnBP and Gln more than five decades ago (**Berntsson et al., 2010**), an effective on-rate constant of about $10^8$ M$^{-1}$s$^{-1}$ has been obtained from numerical fits of stopped-flow relaxation curves at concentration ratios of 1:1 and 2:1 of Gln and GlnBP. For a plausible range $3 \cdot 10^7$ M$^{-1}$s$^{-1}$ < $k_{on}$ < $10^8$ M$^{-1}$s$^{-1}$ of on-rate constants and $K_d$ values of 10–20 nM from different methods (ITC, smFRET, SPR), we obtain 0.3 s$^{-1}$ < $k_{off}$ < 2 s$^{-1}$ as a range for the effective off-rate constant $k_{off} = K_d\, k_{on}$. Together with **Equation 5**, our smFRET results with lower bounds of 100 s$^{-1}$ for the conformational

exchange rates $k_e$ and $k_r$ (corresponding to timescales >10ms) and an upper bound of about 10% for the relative probability $P_{OL} = k_e / (k_e + k_r)$ of the open liganded conformation OL among the two liganded conformations of GlnBP lead to

$$k_{off} = 0.3 \text{ to } 2 \text{ s}^{-1} < k_e < k_r < 100 \text{ s}^{-1} \tag{7}$$

This equation shows that the IF pathway is compatible with our results. *Equation 7* in turn results in a lower bound for $P_{OL}$ of about 0.3 to 2%. We thus consider IF to be the simplest model that correctly describes the ligand-binding mechanism in GlnBP in light of the data and simulations presented here, but clearly state that CS remains possible in case that exchange between the open and closed states in GlnBP is very fast. Another argument to support IF is the notion that the open conformation is more likely to bind substrate than the closed one, based on steric arguments (see Appendix 3). A potential improvement in our argumentation would be to include relaxation kinetics (*Schröder et al., 2023*) without the mass transport limitations in SPR, which is particularly relevant for small ligand molecules. Thus, stopped-flow (FRET) experiments, which have already been used in the 1970s for binding-rate determination in GlnBP (*Tims and Widom, 2007*), would be a more direct approach that could complement smFRET data and might lead to more robust conclusions, as presented above.

What implications do our results and the proposed integrative strategy for determining (or excluding) ligand-binding mechanisms have for other protein systems? Generally, we encourage the use of similar strategies for other biomacromolecular systems, and revisiting various SBP systems (and their binding mechanisms). This is relevant since there are many findings and controversial interpretations whenever intrinsic conformational motion or closed-unliganded conformations were identified for MalE (*Bucher et al., 2011a*; *Bucher et al., 2011b*; *Stockner et al., 2005*), HisJ (*Jayanthi et al., 2020*), GGBP (*Flocco and Mowbray, 1994*; *Ortega et al., 2012*; *Luck and Falke, 1991*; *Careaga et al., 1995*), FBP (*Atilgan and Atilgan, 2009*), ChoX (*Oswald et al., 2008*), and the LAO protein (*Oh et al., 1994*). Also, the advent of single-molecule approaches, such as nanopore-recordings (*Van Meervelt et al., 2017*) and single-molecule Förster-resonance energy transfer (smFRET) (*Kim et al., 2013*; *Seo et al., 2014*; *Gouridis et al., 2015*; *Husada et al., 2015*; *Feng et al., 2016*; *Zhang et al., 2020*; *de Boer et al., 2019b*; *Ploetz et al., 2021*), provided a large pool of data for various ABC transporter-related SBPs (*de Boer et al., 2019b*; *Gouridis et al., 2021*) with a wide range of distinct ligands, such as metal ions (*de Boer et al., 2019b*; *Luo et al., 2021*), osmolytes (*de Boer et al., 2019b*; *Tassis et al., 2021*), amino acids (*Gouridis et al., 2015*; *Husada et al., 2015*; *Feng et al., 2016*; *Zhang et al., 2020*; *de Boer et al., 2019b*; *Ploetz et al., 2021*), peptides (*de Boer et al., 2019b*), sugars (*Mächtel et al., 2019*; *de Boer et al., 2019b*; *Harris et al., 2022*; *Gebhardt et al., 2021*; *Peter et al., 2022*), siderophores (*de Boer et al., 2019a*), and other small molecules (*Peter et al., 2021*) – for most of which additional kinetic data is required to univocally assign a kinetic ligand-binding mechanism.

While SBPs exhibit somewhat conserved structures, certain members show collective differences in structural key features. For example, type I and type II SBPs differ in their overall core topology and in the composition of their hinge domain with two ß-strands for type II, but three strands for the type I family (*Fukami-Kobayashi et al., 1999*). Applying our strategy in a comparative study could help to reveal how such hinge-domain differences contribute to conformational dynamics, thereby strengthening the link between proteins' secondary structure elements, three-dimensional architecture, and function.

## Materials and methods

All commercially obtained reagents were used as received, unless stated otherwise. The following grades were used: Guanidine hydrochloride (99%, Sigma Aldrich), 1,4-Dithiothreitol (DTT; 99%, ROTH), Thermo Fisher Scientific SnakeSkin TM Dialysis Tubing (Thermo Fisher Scientific, 10K MWCO, 16 mm), Ni$^{2+}$-Sepharose resin (GE Healthcare), Albumin fraction V (BSA), biotin-free, ≥98% (Carl Roth GmbH), Imidazole, ≥99% (Carl Roth GmbH), Isopropyl-β-D-1-thiogalactopyranose (IPTG), ≥99% (Carl Roth GmbH), Kanamycin (Carl Roth GmbH), L-glutamine (Merck KGaA), L-Arginine (Carl Roth GmbH). AF555 (Jena Bioscience, Germany), AF647 (Jena Bioscience, Germany), ATTO 532 (ATTO-TEC, Germany), ATTO 643 (ATTO-TEC, Germany), mPEG3400-silane (abcr, AB111226) and biotin-PEG3400-silane (Laysan Bio Inc), Biotin-NTA (Biotium), Streptavidin (Roth, Germany), Pyranose

oxidase (Sigma Aldrich, Germany), Catalase (Sigma Aldrich, Germany), Glucose (≥99.5% GC, Sigma Aldrich, Germany), Trolox (98%, Sigma Aldrich, Germany), Potassium hydroxide (≥85%, Honeywell, Germany), Acetone (Roth, Germany), Toluene (Roth, Germany).

## Protein expression and purification

Two GlnBP double cysteine variants were generated by site-directed mutagenesis, allowing the insertion of two cysteine residues into GlnBP at positions (V111C – G192C) and (T59C – T130C), separately. *E. coli* BL21-pLysS cells were freshly transformed with the plasmid carrying the coding sequence for GlnBP WT or a GlnBP variant, and grown in 2 L LB medium (100 mg/mL Kanamycin and 50 mg/mL chloramphenicol) at 37 °C under aerobic conditions. At an $OD_{600nm}$ of 0.6–0.8, overexpression of the proteins of interest was induced upon addition of 1 mM IPTG to the culture media. The cells were further grown for 1.5–2.0 hr after induction and then harvested by centrifugation for 20 min at $1529 \times g$ (Beckman, JA10) at 4 °C. All subsequent operations were carried out at 4 °C, and all solutions were stored at 4 °C. Cell pellets from 2 L culture were collected in a 50 mL falcon and resuspended in buffer A (50 mM Tris-HCl, pH 8.0, 1 M KCl, 10 mM imidazole, 10% glycerol) with 1 mM dithiothreitol (DTT) and gently shaken overnight at 4 °C.

Cells were disrupted by sonication (Branson tip sonication; amplitude: 25%; 10 min; 0.5 s on-off pulses; temperature was kept low by the use of an ice-water bath). Centrifugation was used to fractionate the cell lysate (at 4 °C for 30 min at $4416 \times g$, Eppendorf, Centrifuge 5804 R) and at 4 °C for 1 hr for ultracentrifugation ($70,658 \times g$, Beckman, Type 70Ti) in vacuum, and the pellet was discarded. The protein was purified by affinity chromatography using the $Ni^{2+}$-Sepharose fast flow resin (GE Healthcare), pre-equilibrated with 10 column volumes of buffer A containing 1 mM DTT and gravity loaded with the supernatant from the preceding ultra-centrifugation step. The resin-bound protein was washed with 10 column volumes of buffer A containing 1 mM DTT, followed by buffer B containing 1 mM DTT (50 mM Tris-HCl, pH 8.0, KCl 50 mM, imidazole 20 mM, glycerol 10%), and finally eluted in buffer C (50 mM Tris-HCl, pH 8.0, KCl 50 mM, imidazole 250 mM, glycerol 10%) with 1 mM DTT. The eluted sample was concentrated (Vivaspin6 columns, 10 kDa MWCO, 6 mg/mL), dialyzed against PBS buffer supplemented with 1 mM DTT, and stirred gently at 4 °C overnight. SDS-PAGE was used to quantify the yield of protein overexpression and purification (Coomassie staining). The absorbance at 280 nm was used to estimate the protein concentration (knowing the molar extinction coefficient of GlnBP $\sim 25,900\ M^{-1}\ cm^{-1}$). The protein was then split into aliquots and kept at –20 °C. All proteins were further purified using SEC (ÄKTA pure system, Superdex 75 Increase 10/300 GL, GE Healthcare). The purified protein was split into aliquots and stored at –80 °C prior to the measurements.

## Unfolding and refolding process of GlnBP WT and GlnBP variants

The stock concentrations of GlnBP variants were estimated at about 6 mg/mL. Each GlnBP variant was thawed from –80 °C, then the protein was diluted to a final concentration of 3–4 µM (final volume of ~20 mL) in the unfolding buffer (PBS buffer) containing 6 M guanidine hydrochloride (GndHCl). Subsequently, the solution was incubated for 3 hr under gentle stirring at ambient temperature. Next, the unfolded GlnBP variants were centrifuged ($3046 \times g$, 30 min at 4 °C) to remove insoluble aggregates which could act as nuclei to trigger aggregation during the refolding process. A Snakeskin TM dialysis membrane was prepared (pre-cooled at 4 °C and soaked in refolding buffer - PBS buffer with 1 mM DTT, pH 7.4 - for 2 min). The GlnBP variants were transferred into the dialysis tubing, which were sealed tightly afterwards by double-knots and clips at each end. The unfolded GlnBP variant was refolded by a two-step dialysis, in the presence of a total 200-fold excess of refolding buffer. First, each protein was dialyzed against 2 L refolding buffer overnight under gentle stirring at 4 °C. Then, buffer was exchanged with an additional 2 L refolding buffer for another day at 4 °C. The refolded protein was then concentrated from 20 mL to a final 500 µL (Vivaspin 10 kDa MWCO; $3000 \times g$, 15 min at 4 °C) and further purified by SEC (ÄKTA pure system, Superdex-75 Increase 10/300 GL, GE Healthcare). The unfolding and refolding process for GlnBP WT was conducted under the same conditions as described for the GlnBP variants.

## Isothermal titration calorimetry (ITC) measurements

The ITC measurements were performed in a MicroCal PEAQ-ITC isothermal titration calorimeter (Malvern Instruments). The prediction ITC software 'MicroCal PEAQ-ITC Control' was employed for

designing and conducting the experiments. Once the $K_d$ value and the binding stoichiometry (N) were assigned as predefined values, the concentration of both the protein and the titrant (ligand) stock solutions could be calculated by the 'design-experiment' function on the software to get an optimal sigmoidal one-site binding curve. GlnBP concentration was assessed using the Nanophotometer (N60 Touch, Implen GmbH) with at least three reading repeats to get accurate determinations of concentration values. For all ITC measurements, the temperature was set at 25 °C with stirring speed at 750 rev/min. The GlnBPs solution (10 μM in PBS buffer pH 7.4, 300 μL) was manually loaded into the sample cell. The titrant (L-glutamine, 100 μM in PBS buffer, pH 7.4) was automatically loaded into the titration syringe and injected in the sample cell with a titration speed of 2 μL every 150 s and a total of 19 injections. As a control experiment, L-glutamine was titrated into the sample cell containing PBS buffer without GlnBPs. All the titration data were analyzed using the MicroCal PEAQ-ITC Analysis Software.

## Surface plasmon resonance spectroscopy (SPR) and data analysis

SPR assays were performed on a Biacore T200 (Cytiva) using a CM5 Series S carboxymethyl dextran sensor chip coated with His-antibodies from the Biacore His-capture kit (Cytiva). Briefly, the chips were equilibrated with running buffer until the dextran matrix was swollen. Afterwards, two flow cells of the sensor chip were activated with a 1:1 mixture of N-ethyl-N-(3-dimethylaminopropyl) carbodiimide hydrochloride and N-hydroxysuccinimide according to the standard amine coupling protocol. A final concentration of 50 μg/mL anti-histidine antibody in 10 mM acetate buffer pH 4.5 was loaded onto both flow cells using a contact time of 420 s for gaining a density of approximately 10,000 resonance units (RU) on the surface. By injection of 1 M ethanolamine/HCl pH 8.0, free binding sites of the flow cells were saturated. Preparation of chip surfaces was carried out at a flow rate of 10 μL/min. All experiments were carried out at a constant temperature of 25 °C using PBS buffer (0.01 M phosphate buffer, 2.7 mM KCl, 0.137 M NaCl, pH 7.4) supplemented with 0.05% (v/v) detergent P20 as running buffer.

For interaction analysis, GlnBP-6His (1.5 μM) was captured onto one flow cell using a contact time of 240 s at a constant flow rate of 10 μL/min. This resulted in a capture density of approximately 1200 RU of GlnBP-6His. Eight different concentrations of glutamine (7.8, 15.6, 31.25, 62.5, 125, 250, 500, and 1000 nM) were injected onto both flow cells using an association time of 50 s and a dissociation time of 360 s. The flow rate was kept constant at 30 μL/min. As control, running buffer was injected. The chip was regenerated after each cycle by removing GlnBP-6His completely from the surface using 10 mM glycine pH 1.5 for 60 s at a flow rate of 30 μL/min.

Sensorgrams were recorded using the Biacore T200 Control software 2.0.2. The surface of flow cell 1 was not coated with GlnBP-6His and used to obtain blank sensorgrams for subtraction of the bulk refractive index background with the Biacore T200 Evaluation software 3.1. The referenced sensorgrams were normalized to a baseline of 0. Peaks in the sensorgrams at the beginning and the end of the injection are due to the run-time difference between the flow cells for each chip.

In total, 26 SPR sensorgrams in three sets of measurements were recorded. To correct for remaining drift in the sensorgrams, the initial 60 s of the sensorgrams prior to Gln injection and the last 100 s of the dissociation phase were first fitted with an exponential function, which was subtracted from the sensorgrams. The drift-corrected sensorgrams were fitted to the reaction scheme of *Equation 1* based on the differential equations (*Schuck and Zhao, 2010*; *Myszka et al., 1998*).

$$\frac{d\,[\mathrm{L}]_{\mathrm{surf}}}{dt} = k_t \left([\mathrm{L}]_{\mathrm{bulk}} - [\mathrm{L}]_{\mathrm{surf}}\right) - \frac{d\,[\mathrm{PL}]}{dt}$$

$$\frac{d\,[\mathrm{PL}]}{dt} = k_{\mathrm{on}}\,[\mathrm{L}]_{\mathrm{surf}}\left([\mathrm{P}]_{tot} - [\mathrm{PL}]\right) - k_{\mathrm{off}}\,[\mathrm{PL}]$$

where $[\mathrm{L}]_{\mathrm{bulk}} = [\mathrm{Gln}]$ and $[\mathrm{L}]_{\mathrm{surf}}$ are the free glutamine concentrations in the bulk flow and at the sensor surface, $[\mathrm{P}]_{\mathrm{tot}}$ is the total concentration of surface-immobilized protein, and $[\mathrm{PL}]$ is the concentration of bound protein complexes. Conversion to the SPR binding response $r$ via $[\mathrm{PL}] = \alpha\,r$ and $[\mathrm{P}]_{\mathrm{tot}} = \alpha\,r_{max}$ leads to fit results for the binding rate constants that are insensitive to the (unknown) conversion factor α, which can be understood from the fact that the quasi-steady-approximation $d\,[\mathrm{L}]_{\mathrm{surf}}/dt \approx 0$ holds for SPR setups (*Schuck and Zhao, 2010*; *Myszka et al., 1998*). The association phases of the sensorgrams were fitted with initial conditions $[\mathrm{L}]_{\mathrm{surf}} = 0$ and $r = 0$ and fit parameters $k_t$ and $r_{max}$ at different values of $k_{\mathrm{on}}$ after substitution of $k_{\mathrm{off}}$ by $K_d k_{\mathrm{on}}$. Prior to these fits with fixed $k_{\mathrm{on}}$, a remaining small vertical off-set of the sensorgrams was determined as an additional fit parameter

in fits with unconstrained, large $k_{on}$ and subtracted from the sensorgrams. The first 50 s of the dissociation phases were fitted with a single fit parameter $k_t$ for the initial conditions $[L]_{surf} = [L]_{bulk}$ and $r = r_{max}/\left(1 + K_d/[L]_{bulk}\right)$, with $r_{max}$ determined from fits of the association phase of the sensorgram for unconstrained, large $k_{on}$. Background-corrected sensorgrams that do not reach binding equilibrium in the association phase (because of small [Gln]), still show marked drifts in binding equilibrium, or do not resolve the initial increase of the binding signal of the association phase (because of large [Gln]) were discarded, which leads to the 13 sensorgrams of *Figure 6B,C*, *Figure 6—figure supplement 1* with fit results for $\alpha$=1 μM/RU. Fits with, for example, $\alpha$=1 mM/RU (not shown) lead to practically identical results. All fits were conducted with Mathematica 13 based on the functions ParametricNDSolveValue to obtain numerical solutions of the differential equations and NonlinearModelFit for fitting parameters of these solutions.

## Protein labeling

The refolded GlnBP(111 C-192C) and GlnBP(59 C-130C) variants were labeled with commercial maleimide derivatives of AF555/AF647 or ATTO 532/ATTO 643 (*Gebhardt et al., 2021*), and then purified by SEC. The chromatogram of refolded GlnBP(111 C-192C) labeled with AF555/AF647 is shown in *Figure 2B*, and those of all other variants and dye labeling combinations are displayed in *Figure 2—figure supplement 2*. First, the His-tagged protein was incubated in 10 mM DTT in PBS buffer for 30 min to reduce all oxidized cysteine residues. Subsequently, the protein was diluted 10 times with PBS buffer and immobilized on a Nickel Sepharose 6 Fast Flow resin (GE Healthcare). The resin was washed extensively with milliQ water followed by PBS buffer pH 7.4. To remove the excess of DTT, the resin was washed with PBS buffer. The protein was left on the resin and incubated overnight at 4 °C with 5–10 times molar dye excess in PBS buffer pH 7.4. Subsequently, the unreacted fluorophores were removed by washing the resin with 6 mL of PBS buffer. Bound proteins were eluted with 800 μL of elution buffer (PBS buffer, pH 7.4, 400 mM Imidazole). The labeled protein was further purified by SEC (ÄKTA pure, Superdex-75 Increase 10/300 GL, GE Healthcare) to eliminate remaining fluorophores and remove other contaminants and soluble aggregates. The selected elution fractions were used without further treatment for smFRET experiments as described below. In general, all experiments were carried out at room temperature using 25–50 pM of double-labeled GlnBP protein in PBS buffer (pH 7.4). Titration experiments were performed by adding specific concentrations of ligand (glutamine) to the buffer.

## smFRET experiments with μsALEX

Single-molecule μsALEX experiments were carried out at room temperature on a custom-built confocal microscope. In short, alternating excitation light (50 μs period) was provided by two diode lasers operating at 532 nm (OBIS 532–100-LS, Coherent, USA) and 640 nm (OBIS 640–100 LX, Coherent, USA). Both lasers were combined by coupling them into a polarization maintaining single-mode fiber (P3-488PM-FC-2, Thorlabs, USA) and subsequently guided into the microscope objective (UplanSApo 60 X/1.20 W, Olympus, Germany) via a dual-edge dichroic mirror (ZT532/640rpc, Chroma, USA). In general, the 532 and 640 nm diode lasers operated at 60 and 25 μW, respectively (measured at the back aperture of the objective), unless stated otherwise. Fluorescence light was collected by the same objective, focused onto a 50 μm pinhole and separated into two spectral channels (donor and acceptor fluorescence) by a dichroic beamsplitter (H643 LPXR, AHF, Germany). Fluorescence emission was collected by two avalanche photodiodes (SPCM-AQRH-64, Excelitas) after additional filtering (donor channel: BrightLine HC 582/75 and acceptor channel: Longpass 647 LP Edge Basic, both from Semrock, USA). The detector outputs were recorded via an NI-Card (PCI-6602, National Instruments, USA) using a custom-written LabView program.

## smFRET data analysis (μsALEX)

Data analysis for μsALEX was performed using an in-house written software package as previously described (*Gouridis et al., 2015*). Three relevant photon streams were extracted from the recorded data based on the alternation period, corresponding to donor-based donor emission F(DD), donor-based acceptor emission F(DA), and acceptor-based acceptor emission F(AA). Bursts from single-molecules were identified using published procedures (*Nir et al., 2006*) based on an all-photon-burst-search

algorithm with a threshold of 15, a time window of 500 µs, and a minimum total photon number (F(DD)+D(DA)+F(AA)) of 150, unless stated otherwise in the figure caption.

For each fluorescence burst, the stoichiometries S* and apparent FRET efficiencies E* were calculated and then presented for all bursts yielding a two-dimensional (2D) histogram. Uncorrected apparent FRET efficiency, E*, monitors the proximity between the two fluorophores and is calculated according to E*=F(DA)/(F(DD)+F(DA)). Apparent stoichiometry, S*, is defined as the ratio between the overall fluorescence intensity during the green excitation period over the total fluorescence intensity during both green and red periods and describes the ratio of donor-to-acceptor fluorophores in the sample: S*=(F(DD)+F(DA))/(F(DD)+F(DA)+F(AA)). Collecting the E* and S* values of all detected bursts into a 2D E*-S* histogram yielded subpopulations that can be separated according to their E*- and S*-values. The 2D histograms were fitted using a 2D Gaussian function, yielding the mean apparent FRET efficiency and its SD or width of the distribution. µsALEX assists in sorting single molecules based on their donor/acceptor dye brightness ratio (stoichiometry S*) and uncorrected mean FRET efficiency (apparent FRET E*), which can be related to the mean inter-dye distance (*Gebhardt et al., 2021*; *Lee et al., 2005*).

Analysis with mpH²MM was conducted as described previously by the Lerner lab (*Harris et al., 2022*). In short, the FRET Bursts software (*Ingargiola et al., 2016*) was used for detecting single-molecule photon bursts using the DCBS (*Nir et al., 2006*) AND-gate algorithm with a sliding window of m=10 photons searching for instances with an instantaneous photon rate of at least *F*=6 times the background rate. Afterwards, bursts of such consecutive photons were filtered to have at least 50 photons originating from donor excitation and at least 50 photons originating from acceptor excitation. In the data analysis, the photon stream was then divided into photon streams of different bursts, and a time shift was applied to the acceptor excitation originating photons stream so that their arrival time range overlapped with that of donor excitation originating photon streams. Optimizations were conducted with state models of increasing numbers of states, and the *Viterbi* algorithm was employed for calculating the integrated complete likelihood (ICL). Optimizing for larger numbers of states ceased once the ICL ceased to decrease between successively larger state models. Optimized models were manually examined, and the optimal state model selected considering the ICL and the reasonableness of the model given prior knowledge based on transition rates and the E* and S* values of the states. After selection of the most-likely state model, the corresponding most-likely state-path determined by the *Viterbi* algorithm was used to segment bursts into dwells and to classify bursts by which states were present within each burst.

To support the idea that apo and holo states in solution match with that of the crystal structure, we performed a quantitative comparison of inter-dye distances calculated from dye accessible volumes (AV) on structural models of apo and holo protein, and those derived from the experimental smFRET results. For dye AV calculations, we used the FPS method, established by the Seidel lab (*Kalinin et al., 2012*; *Figure 2—figure supplement 1*). The experimental data were corrected for setup-dependent parameters according to *Hellenkamp et al., 2018*; *Peter et al., 2022* to obtain accurate FRET values from µsALEX data. Using a Förster distance of 5.2 nm for AF555/AF647, we found good agreement, that is 0.3–0.5 nm deviations (and 1.0 nm in one case) between the calculated and experimentally derived inter-dye distances for both mutants (*Figure 2—figure supplement 1*).

## smFRET measurements with MFD-PIE and burst-wise FCS analysis

Solution-based smFRET experiments were performed on a home-built dual-color confocal microscope that combines multiparameter fluorescence detection (MFD) with pulsed interleaved excitation (PIE; *Kudryavtsev et al., 2012*). MFD-PIE experiments have been described in detail previously (*Dahiya et al., 2019*). With MFD-PIE, it is possible to extract FRET efficiency, stoichiometry, fluorescence lifetime, and fluorescence anisotropy information from each single-molecule burst. Correction factors including direct acceptor excitation (α), spectral crosstalk (β), and detection correction factor (γ) are also accounted for reporting accurate the FRET efficiency values (*Agam et al., 2023*). The accurate FRET efficiency (E) can be determined from:

$$E = \frac{F_{GR} - \alpha F_{RR} - \beta F_{GG}}{F_{GR} - \alpha F_{RR} - \beta F_{GG} + \gamma F_{GG}}$$

where $F_{GG}$, $F_{GR}$, and $F_{RR}$ are background-corrected fluorescence signals detected in green/donor (G), red/acceptor (R) after donor excitation and acceptor channels, respectively.

Alternatively, the use of picosecond pulsed lasers and time-correlated single photon counting (TCSPC) electronics enables calculating FRET efficiencies from the quenching of the donor in the presence of acceptor. According to the formula:

$$E = 1 - \frac{\tau_{D(A)}}{\tau_{D(0)}}$$

$\tau_{D(A)}$ is the fluorescence lifetime of the donor in the presence of acceptor and $\tau_{D(0)}$ is the fluorescence lifetime of the donor only species. Static species can be observed on the theoretical static FRET line, which is a linear relation between E and $\tau_{D(A)}$. Sub-ms conformational dynamics can also be identified and judged by observing the right-shifted populations from the static FRET line.

For the measurements here, 100 pM of GlnBP labeled with ATTO 532 and ATTO 643 was placed on a BSA-passivated LabTek chamber and measured for 2 hours. The sample was excited with 532 and 640 nm pulsed lasers with a repetition rate of 26.6 MHz and laser powers of 45 and 23 µW (measured at the back aperture of the objective), respectively.

Burst-wise FCS analysis is an alternative approach to observe sub-ms conformational dynamics. In this approach, donor (DD) and acceptor (AA) signals detected from single-molecule events are cross-correlated. Thus, fluctuations in the FRET efficiencies appear as an anti-correlated signal in the donor-acceptor fluorescence cross-correlation function. Bursts with sufficient photons detected in both the donor and acceptor channels were selected. A time window of 50 ms was applied around each burst. If another burst was detected within this time window, both were eliminated to ensure correlation functions that are specific to the selected bursts. All the above-mentioned data analysis was done using the PIE analysis with Matlab (PAM) software package (*Schrimpf et al., 2018*).

## Surface immobilization of DNA and GlnBP(111C-192C)

Biotin-streptavidin interaction was used to immobilize tagged proteins and labeled DNA on a PEG-functionalized coverslip for single molecule studies. The protein-his-tag and a biotin-NTA chelated with $Ni^{2+}$ were used to mark GlnBP(111 C-192C) labeled with maleimide modified derivatives of ATTO 532/ATTO 643, whilst DNA labeled with Cy3B/ATTO 647 N was directly tagged with a biotin. To prepare a functionalized glass surface, cover slides (1.5 H Marienfeld Superior) were first sonicated in MQ water for 30 min. The slides were rinsed three times with MQ water, sonicated for 30 min in HPLC-grade acetone, rinsed three times with MQ water again. Then, the slides were sonicated with 1 M KOH for 30 min, rinsed three times with MQ water, and dried with a stream of nitrogen air. To remove any organic material left on the surface, the cover slides were plasma-cleaned for 15 min with oxygen. To create a mPEG/biotin–coated surface, the slides were immediately incubated in a 99:1 solution of mPEG3400-silane (abcr, AB111226) and biotin-PEG3400-silane (Laysan Bio Inc) in a Toluene solution overnight at 55 °C. After incubation, the slides were sonicated (10 min in ethanol, 10 min in MQ water), dried under nitrogen stream, and kept under vacuum. Prior to TIRF experiments, each slide was incubated with a 0.2 mg/mL streptavidin in PBS solution for 10 min utilizing Ibidi sticky-slide (18 well) for single molecule studies. PBS buffer pH 7.4 was used to wash away the unbound excess of streptavidin. For GlnBP(111 C-192C) immobilization, 20 nM biotin-NTA (QIAGEN) was charged with 50 nM $Ni^{2+}$ and incubated on the slide for 10 min before rinsing away the unbound excess biotin-NTA and $Ni^{2+}$ with PBS (this step was omitted for the labeled DNA samples). GlnBP(111 C-192C) at 0.8 nM and dsDNA at 0.04 nM were incubated for 5 and 1 min, respectively. For single-molecule data collection, imaging buffer (PBS, pH 7.4) containing 2 mM Trolox was used. For dsDNA, we used PBS buffer in combination with an oxygen scavenging system (pyranose oxidase at 3 U/mL, catalase at final concentration of 90 U/mL, and 40 mM glucose). After that, the chambers were sealed with Silicone Isolators Sheet Material (Grace Bio-labs). All the single-molecule investigations were done at room temperature.

## smFRET measurements with TIRF microscopy including data analysis

Single-molecule TIRF measurements were conducted on a homebuilt microscope using an Olympus iX71 inverted microscope body. Light from a 532 nm continuous wave laser (532 nm OBIS, Coherent) was transmitted off-axis onto the back-focal plane of a microscope objective (UAPON TIRF 100X1.49 NA,

Olympus) via a dual-band dichroic beamsplitter (TIRF Dual Line Beamsplitter zt532/640rpc, AHF Analysetechnik) to generate total internal reflection at the glass-water interface. Fluorescent emission was then split spectrally using a Dual View System (DV2, Photometrics) equipped with a dichroic beamsplitter (zt640rdc, AHF Analysetechnik). The two emission channels were then spectrally filtered using emission filters (582/75 Brightline HC and 731/137 BrightLine HC respectively, both AHF Analysetechnik). Image series were acquired using an EMCCD camera (C9100-13, Hamamatsu) in combination with the μManager (*Ingargiola et al., 2016*) software. The iSMS (*Kalinin et al., 2012*) software was used to retrieve and calculate traces of the donor and acceptor fluorescence intensity from consecutive fluorescent images.

## MD simulations

Starting point of our atomistic simulations with the AMBER20 software package (*Case et al., 2020*) and the ff14SB force field parameters (*Maier et al., 2015*) was the ligand-bound crystal structure with PDB identifier 1WDN. The protein state of the titratable amino acids and including ligand was determined with the software PROPKA3 (*Olsson et al., 2011*). The protein, with and without ligand, was solvated in explicit TIP3P water in an octahedral simulation box with a minimum distance of 15 Å of protein atoms to the box boundaries at a salt concentration of 150 mM. The two simulation systems with and without ligand were carefully relaxed in nine steps according to the AMBER tutorial 'Relaxation of explicit water systems' (see https://ambermd.org/tutorials/). Production simulations starting from the system conformations obtained after relaxation were performed with the AMBER20 software implementation for GPUs (*Case et al., 2020*; *Salomon-Ferrer et al., 2013*) with a time step of 4 fs after hydrogen-mass repartitioning (*Hopkins et al., 2015*). In these simulations, the temperature was kept at 300 K using a Langevin thermostat with a collision frequency of 1 ps$^{-1}$ and the pressure was kept at 1 bar with the Berendsen barostat. To observe unbinding events on the microsecond timescale accessible in our simulations, we reduced all interactions between the protein and the ligand by 16% by rescaling the partial charges and ε parameters of the ligand atoms with the commands *change charge* and *changeLJSingleType* of the program ParmEd implemented in Amber.

## Acknowledgements

This work was financed by the European Commission (ERC-STG 638536 – SM-IMPORT to TC), Deutsche Forschungsgemeinschaft (GRK2062, project C03 to TC); SFB863, project A13 to TC and Sachbeihilfe CO 879/4–1 to TC, Project 449926427 to KJ, SFB1035 (201302640, project A11 to DCL), the Bundesministerium für Bildung und Forschung (KMU grant "quantumFRET" to TC), the Israel Science Foundation (grants 556/22 and 3565/20 to EL), NIH (grant R01 GM130942 to EL as subaward) and the Center for Nanoscience (CeNS). This work was also supported by the Federal Ministry of Education and Research (BMBF) and the Free State of Bavaria under the Excellence Strategy of the Federal Government and the Länder through the ONE MUNICH Project Munich Multiscale Biofabrication (to DCL). ZH acknowledges a PhD scholarship from the Chinese Scholarship Council (CSC), PDH the 2022–2023 Zuckerman STEM postdoctoral fellowship and PC and NZ postdoctoral fellowships from the Alexander von Humboldt foundation. TW acknowledges support by the Max Planck Society. We thank E Cabrita for advice on the interpretation of NMR data.

## Additional information

### Competing interests

Thorben Cordes: is a scientific co-founder and share-holder of FluoBrick Solutions GmbH a company that distributes fluorescence microscopy and spectroscopy instruments. The other authors declare that no competing interests exist.

### Funding

| Funder | Grant reference number | Author |
|---|---|---|
| European Commission | 10.3030/638536 | Thorben Cordes |

| Funder | Grant reference number | Author |
| --- | --- | --- |
| Deutsche Forschungsgemeinschaft | CO 879/4-1 | Thorben Cordes |
| Deutsche Forschungsgemeinschaft | GRK2062 | Thorben Cordes |
| Deutsche Forschungsgemeinschaft | SFB863 | Thorben Cordes |
| Alexander von Humboldt Foundation | Postdoctoral fellowship | Pazit Con Niels Zijlstra |
| Israel Science Foundation | 556/22 | Eitan Lerner |
| Israel Science Foundation | 3565/20 | Eitan Lerner |
| National Institute of Health | R01 GM130942 | Eitan Lerner |
| Max Planck Society | | Thomas R Weikl |
| the Center for Nanosicence (CeNS) | | Eitan Lerner |
| Chinese Scholarship Council | PhD scholarship | Zhongying Han |
| Zuckerman STEM | 2022–2023 Postdoctoral fellowship | Paul David Harris |
| Bundesministerium für Forschung, Technologie und Raumfahrt | | Don C Lamb |
| Free State of Bavaria | | Don C Lamb |
| Deutsche Forschungsgemeinschaft | SFB1035 | Don C Lamb |
| Deutsche Forschungsgemeinschaft | 449926427 | Kirsten Jung |
| Bundesministerium für Forschung, Technologie und Raumfahrt | quantumFRET | Thorben Cordes |

The funders had no role in study design, data collection and interpretation, or the decision to submit the work for publication. Open access funding provided by the Max Planck Society.

## Author contributions

Zhongying Han, Conceptualization, Data curation, Formal analysis, Validation, Investigation, Visualization, Methodology, Writing – original draft; Sabrina Panhans, Marija Ram, Resources, Investigation, Methodology, Writing – review and editing; Sophie Brameyer, Anna Herr, Alessandra Narducci, Investigation, Methodology, Writing – review and editing; Ecenaz Bilgen, Paul David Harris, Software, Investigation; Michael Isselstein, Eitan Lerner, Software, Investigation, Writing – review and editing; Oliver Brix, Investigation; Pazit Con, Kirsten Jung, Writing – review and editing; Don C Lamb, Investigation, Writing – review and editing; Douglas Griffith, Conceptualization, Resources, Investigation, Writing – review and editing; Thomas R Weikl, Conceptualization, Data curation, Software, Formal analysis, Supervision, Funding acquisition, Writing – original draft, Writing – review and editing; Niels Zijlstra, Conceptualization, Formal analysis, Supervision, Writing – review and editing; Thorben Cordes, Conceptualization, Formal analysis, Supervision, Funding acquisition, Writing – original draft, Project administration, Writing – review and editing

## Author ORCIDs

Ecenaz Bilgen ⓘ https://orcid.org/0000-0003-4919-5104
Don C Lamb ⓘ https://orcid.org/0000-0002-0232-1903
Eitan Lerner ⓘ https://orcid.org/0000-0002-3791-5277
Thomas R Weikl ⓘ https://orcid.org/0000-0002-0911-5328
Thorben Cordes ⓘ https://orcid.org/0000-0002-8598-5499

Reviewer #1 (Public review): https://doi.org/10.7554/eLife.95304.3.sa1
Reviewer #2 (Public review): https://doi.org/10.7554/eLife.95304.3.sa2
Author response https://doi.org/10.7554/eLife.95304.3.sa3

## Additional files

### Supplementary files
MDAR checklist

### Data availability
Data available on Zenodo and Edmond.

The following datasets were generated:

| Author(s) | Year | Dataset title | Dataset URL | Database and Identifier |
|---|---|---|---|---|
| Han Z, Cordes T | 2026 | Dissecting Mechanisms of Ligand Binding and Conformational Changes in the Glutamine-Binding Protein | https://doi.org/10.5281/zenodo.20027916 | Zenodo, 10.5281/zenodo.20027916 |
| Weikl T | 2026 | Atomistic molecular dynamics trajectories of the glutamine-binding protein GlnBP | https://doi.org/10.17617/3.ZVKZPB | Edmond, 10.17617/3.ZVKZPB |

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

## Appendix 1

### Interpretation of mpH$^2$MM analysis

For analysis of within-burst dynamics, we used multi-parameter photon-by-photon hidden Markov modeling (mpH$^2$MM; *Harris et al., 2022*; *Pirchi et al., 2016*) to identify the most-likely state model that describes the experimental results based on how E* and S* values may change within single-molecule bursts. For this analysis, we (i) report the most-likely number of states and their mean E* and S* values (*Figure 4B*, red dots). (ii) We investigate whether molecules traversing the confocal excitation volume are fully static and only in the mid-FRET state or high-FRET state, or whether they undergo dynamic FRET changes including transitions of mid/high-FRET states with photo-blinking dynamics or dark donor or acceptor states (*Figure 4B*). (iii) We finally report on E* and S* values for parts of bursts with dwells in one of the identified states and the rate constants of transitioning between them (*Figure 4B*). These analyses confirm that among the two types of dynamic transitions that influence the burst-based E* and S* values, these are mostly donor or acceptor photo-blinking dynamics between bright and dark states of the fluorophores. Such behavior is irrelevant to understanding the conformational changes in GlnBP but does influence the mean FRET efficiency values if not decoupled. Importantly, no dynamic transitions occur between the mid-FRET and high-FRET states at timescales shorter than 10ms (i.e. with rate constants higher than 100 s$^{-1}$). All measurement conditions show significant photo-blinking dynamics which occur mostly on few ms to sub-millisecond timescales most prominently for the use of AF555/AF647 and the GlnBP(59/130) variant (compare *Figure 4*, *Figure 4—figure supplements 1–3*). Therefore, the blinking dynamics likely account also for the signature of within-burst dynamics shown by BVA (*Figure 4*, *Figure 4—figure supplements 1–3*).

Most importantly, mpH$^2$MM identifies single apo and holo E*-states, which describe the open mid-FRET and closed high-FRET conformations of GlnBP. Only in the presence of low (near K$_D$) concentrations of glutamine two FRET states are identified which interconvert on timescales slower than 10ms. Notably, the mean E* and S* values of the FRET states are slightly dissimilar to the centers of the burst-based E* and S* populations, owing to the effect of the rapid photo-blinking dynamics within bursts, which lead to averaging the E* and S* values of the FRET states with those of the photo-blinked states. Additionally, in the presence of near-K$_D$ concentrations of glutamine, the FRET dynamics occur in the few ms timescale or even slower, which may contribute only slightly to the signature of FRET dynamics in BVA. In conclusion, if intrinsic conformational dynamics existed in apo GlnBP, it could only be between the highly-populated FRET conformation we identify and another conformation that is populated way below the sensitivity of our measurement and analysis (potentially <5–10% populations). Thus, we can conclude that the majority of the conformational dynamics in GlnBP is induced by glutamine, most probably as a result of its binding to GlnBP.

## Appendix 2

### Description of TIRF data acquisition and analysis

At first, we studied a biotin-modified double-stranded DNA (dsDNA), which was labeled with Cy3B (donor) and ATTO 647 N (acceptor) in a 13 bp distance, and used this as a reference sample to allow a direct comparison of µsALEX and TIRF data (*Appendix 2—figure 2*). For this, we immobilized the dsDNA on a PEG-coated glass surface via streptavidin-biotin interactions. We recorded both donor and acceptor fluorescence via a dual-view split on our EMCCD camera with 100ms integration time per frame. With this, we obtained traces that lasted multiple 10 s periods. Since we did not perform millisecond alternation of green-and-red laser excitation, we verified that the sum-signal of the donor and acceptor channel was constant as a function of time for each molecule and discarded traces that did not obey this condition. The dsDNA sample displays an apparent FRET efficiency E* of ~0.64 for in-solution measurements, which agreed well with the analysis of surface-immobilized molecules on the TIRF microscope having a mean E* of 0.62 (*Appendix 2—figure 2A/B*).

Then, we investigated the conformational states and changes of GlnBP(111 C-192C) with the dye pair ATTO 532/ATTO 643, since these showed least photophysical FRET-dynamics (see main text and Appendix 1). To exclude the influence of buffer and other small molecules in TIRF measurements on the conformational state of GlnBP, we initially performed control experiments in µsALEX (*Appendix 2—figure 1*). We found that GlnBP was influenced by the addition of oxygen scavenger cocktails (pyranose oxidase and catalase, POC, and glucose or protocatechuate-dioxygenase, PCD, and 3,4-protocatechuic acid, PCA), resulting in the formation of artificial holo-state GlnBP molecules (*Appendix 2—figure 1E/F*). In TIRF experiments, the effect of oxygen scavenger might have been misinterpreted as intrinsic closing. We consequently proceeded with no oxygen-removal in PBS buffer (pH 7.4) and with 2 mM Trolox as photostabilizer. GlnBP was immobilized by biotin-NTA interactions mediated by Nickel(II). To our surprise, we found very different mean E* values on TIRF in comparison to µsALEX measurements (*Appendix 2—figure 2E/F*). In detail, the mean E* values were much higher on TIRF than on µsALEX (*Appendix 2—figure 2E/F*) in contrast to dsDNA (*Appendix 2—figure 2A/B*). This can be interpreted as an altered conformational state of GlnBP, for example likely caused by protein-glass interactions due to surface-immobilization or interaction of the protein or dyes with the biotin-NTA moiety. Furthermore, the addition of saturating glutamine concentrations did not show the expected behavior of a full shift of the population to a higher-FRET state (*Appendix 2—figure 2F*). Instead, only a small fraction of the population is shifted for both low and saturating glutamine concentrations. At concentrations of glutamine around the $K_d$-value, freely diffusing GlnBP shows a mix of open- and closed states in µsALEX experiments (*Figure 3*). In TIRF, however, we could not identify dynamic transitions (*Appendix 2—figures 2/4*). This finding indicates that a part of the immobilized fluorophore-labeled GlnBP becomes non-functional. Since our protocol deviates from that used in other studies (*Feng et al., 2016*; *Zhang et al., 2020*; *Wu et al., 2022*), we probed whether we could reproduce published data on substrate-binding domains 1 and 2 (SBD1 and SBD2; *de Boer et al., 2019b*). Again, we find a good match between biochemical properties, µsALEX, and the corresponding TIRF data for both proteins (*Appendix 2—figure 5*).

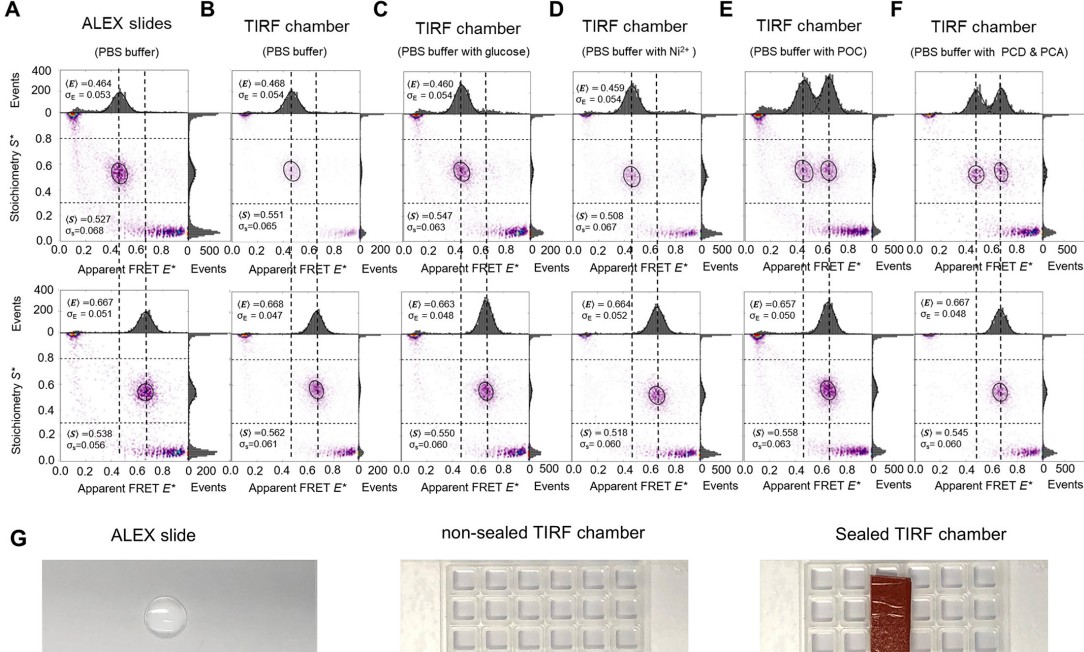

**Appendix 2—figure 1.** Buffer effects on the conformational states of GlnBP(111 C-192C) under various conditions. Due to the high binding affinity of GlnBP for L-glutamine, several control experiments under different conditions were performed to exclude artifacts induced by the reagents present in each set of experiments. The μsALEX experiments of the refolded GlnBP(111 C-192C) double-cysteine variant labeled with LD555/LD655 fluorophore pairs were measured in PBS buffer (pH 7.4) using conventional microscope glass slides (**A**) and using TIRF chamber (**B**). The PBS buffer containing (**C**) 40 mM glucose, (**D**) 50 nM $Ni^{2+}$, (**E**) pyranose oxidase/catalase (POC), and (**F**) protocatechuate-dioxygenase (PCD)/3,4-protocatechuic acid (PCA) was used for the ALEX measurements. (**G**) The conventional glass coverslips used in μsALEX experiments (left figure) and TIRF chambers (sticky-Slide 18 well, Ibidi; non-sealed chambers: middle panel; sealed: right panel) glued on top of PEG-/biotin-PEG-silane microscope glass coverslips used in the TIRF experiments.

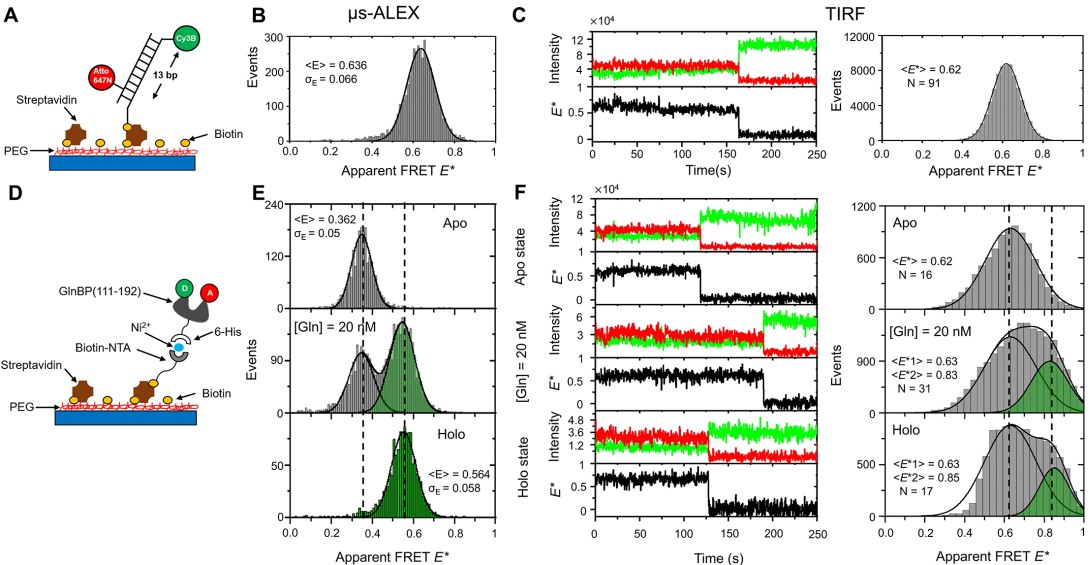

**Appendix 2—figure 2.** Comparing smFRET measurements of biotin-modified dsDNA and GlnBP(111 C-192C) using diffusion-based μsALEX versus TIRF microscopy. (**A**) Schematic view of dsDNA labeled with Cy3B and ATTO 647 N for smFRET characterization on PEGylated coverslips. (**B**) Typical μsALEX-based E*-S* histograms of the biotin-modified dsDNA labeled with Cy3B and ATTO 647 N. (**C**) Representative fluorescence time trace

*Appendix 2—figure 2 continued on next page*

of respective single emitter of the biotin-modified dsDNA sample under continuous wave excitation of ~500 μW at 532 nm and the FRET histograms of all analyzed molecules and the FRET histograms of all measured molecules combined. (**D**) Schematic view of the refolded GlnBP(111 C-192C) labeled with ATTO 532 and ATTO 643 for smFRET characterization. (**E**) Typical μsALEX-based E*-S* histograms of the refolded GlnBP(111 C-192C). (**F**) Representative fluorescence time trace of respective single emitter of the refolded GlnBP(111 C-192C) under continuous wave excitation of ~500 μW at 532 nm and the FRET histograms of all analyzed molecules. Additional data for each condition are shown in *Appendix 2—figure 4*.

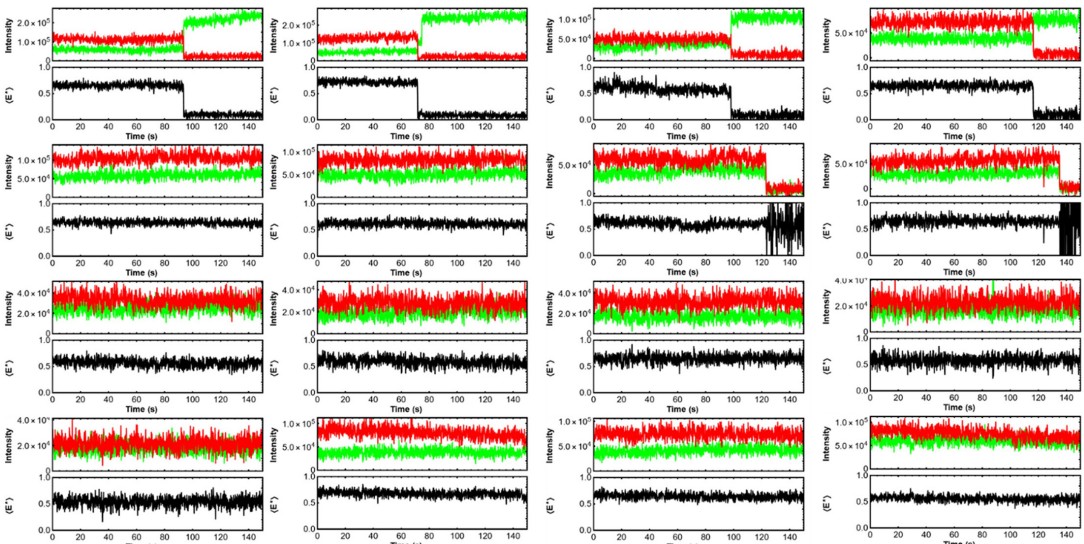

**Appendix 2—figure 3.** Representative fluorescence time traces of respective single emitter of biotin-functionalized DNA labeled by maleimide-modified derivatives Cy3B and ATTO 647 N (13 bp inter-dye distance). All measurements were done in oxygen scavenging buffer (3 U/mL of pyranose oxidase, 90 U/mL of catalase, and 40 mM glucose, PBS buffer, pH 7.4). Laser power: 500 μW.

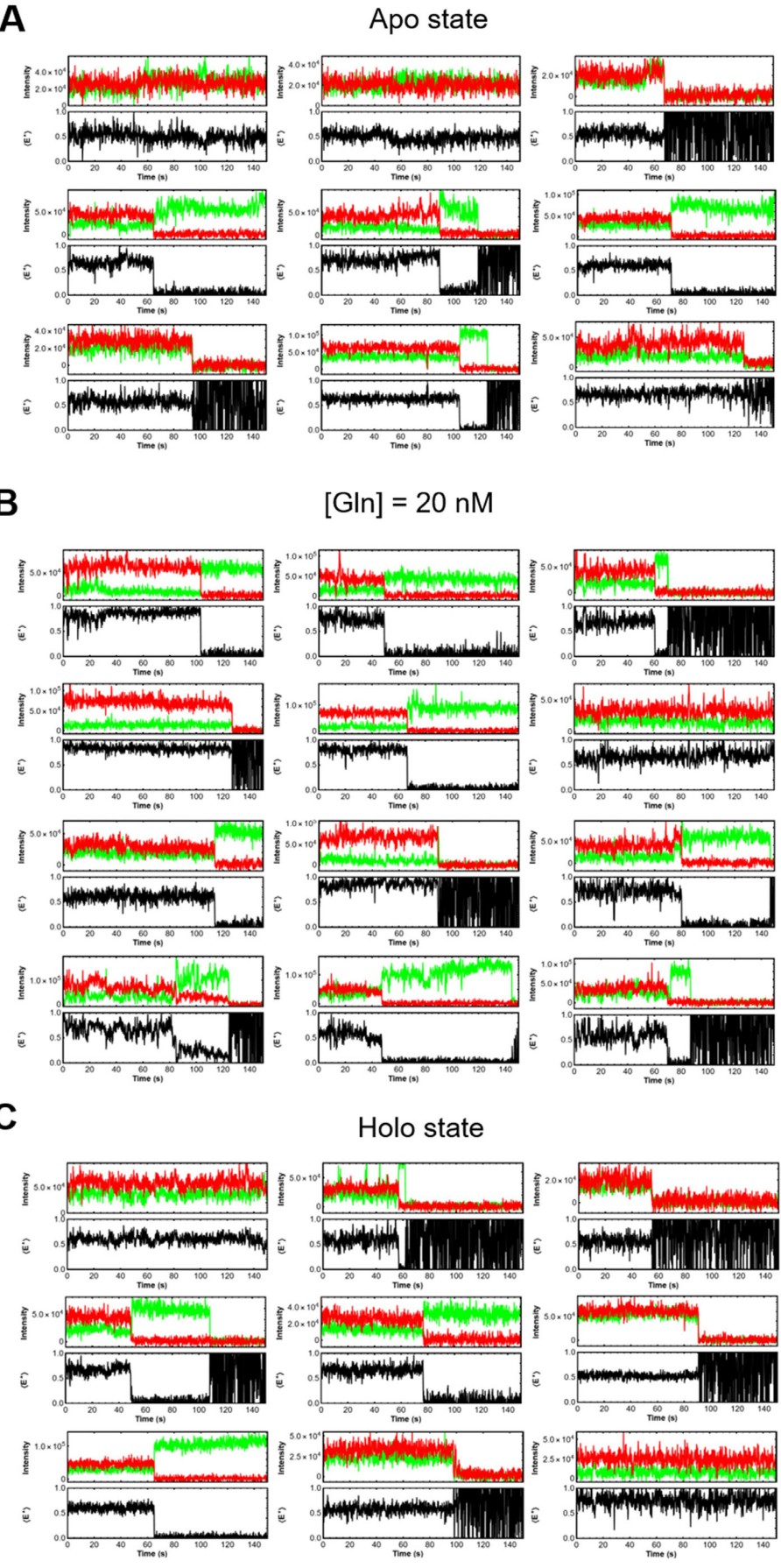

**Appendix 2—figure 4.** Examples of fluorescence time traces of respective single emitter of refolded GlnBP(111 C-192C) labeled by maleimide-modified derivatives ATTO 532 and ATTO 643 under (**A**) apo, (**B**) 20 nM Gln and (**C**) holo conditions. All measurements were done in PBS buffer, pH 7.4, and 2 mM Trolox. Laser power with continuous 532 nm excitation: 200 μW.

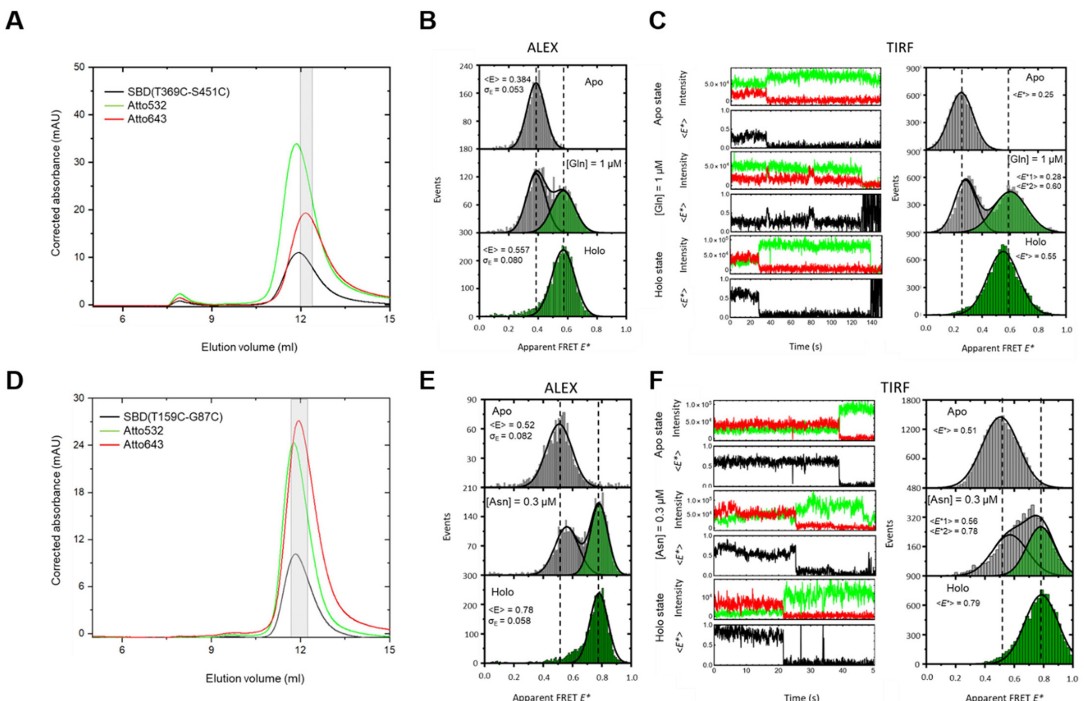

**Appendix 2—figure 5.** Single-molecule studies of reference proteins SBD1 and SBD2. (**A, D**) Size-exclusion chromatography (SEC) of SBD2 (T369C-S451C) and SBD1 (T159C-G87C). The selected fractions (grey-shaded area) were collected and used for the solution-based smFRET measurements. The selected fractions (grey-shaded area) having the best overlap of protein, donor, and acceptor absorption were used. The protein absorption was measured at 280 nm (black curves) and the donor dye (ATTO 532) absorption at 532 nm. The acceptor dye absorption (red lines) was measured at 643 nm for ATTO 643. (**B, E**) Typical μsALEX-based E*-S* histograms of the SBD2(T369C-S451C) and SBD1(T159C-G87C). (**C, F**) Representative fluorescence time trace of respective single emitter of the SBD2(T369C-S451C) and SBD1(T159C-G87C) and the FRET histograms of all measured molecules.

## Appendix 3

### Considerations on the accessibility of the ligand-binding pocket for solvent and ligand in the closed conformation of GlnBP

To describe the expected binding behavior of the substrate glutamine to GlnBP, we performed docking calculations of GlnBP in its open and closed conformations. The GlnBP structure that represents the open conformation is the one reported under pdb code 1GGG (*Hsiao et al., 1996*). The GlnBP structure that represents the closed conformation is the one reported under pdb code 1WDN (*Sun et al., 1998*), with the bound ligand taken out of the file. Then, we used the 3D conformer structure of the ligand to be docked onto the structures of GlnBP. We used the SwissDock web server to perform the docking procedure (*Grosdidier et al., 2011a*; *Grosdidier et al., 2011b*). The results show that (i) while glutamine can dock to many sites on GlnBP, the results that yield the lowest binding free energy are when it docks onto its cognate binding site, both in the open and closed conformations (*Appendix 3—figures 1 and 2*). (ii) The calculated binding free energy of Gln to GlnBP in the optimized docking site leads to a dissociation constant of 20 µM in the open conformation and 230 nM in the closed conformation (*Appendix 3—figure 2*), about two orders of magnitude different. (iii) The higher binding free energy is due to the larger amount of GlnBP residues when the docked glutamine interacts with in the closed conformation relative to in the open conformation. (iv) The binding pocket in GlnBP seems to surround the docked glutamine from all directions (*Appendix 3—figure 1*), which implies that it is less probable that glutamine can access the binding pocket in the closed conformation. Instead, it is more probable that the glutamine reaches its binding site in GlnBP when it is not yet closed.

**1WDN (closed) GlnBP Secondary structure**    **+ main/side chain**    **+ main/side chain + protein surface**    **+ electrostatic potential**

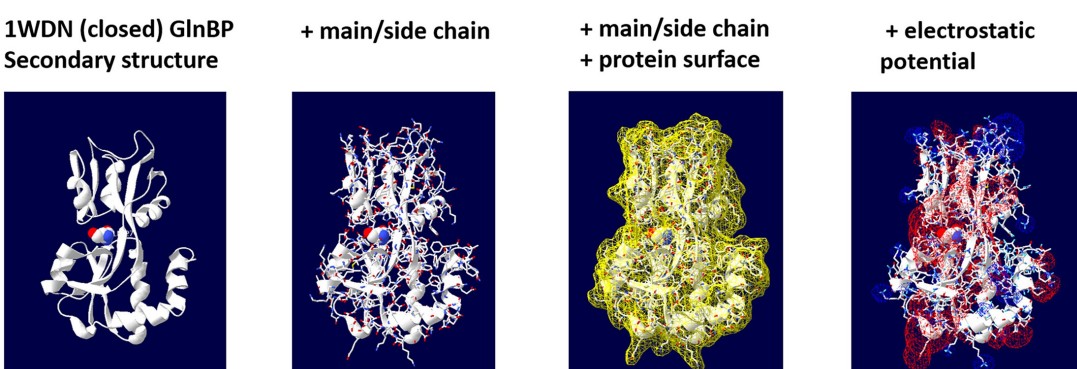

**Appendix 3—figure 1.** The structure of holo GlnBP with optimized docking of glutamine. The figure reports the optimized results of docking glutamine onto the crystal structure of GlnBP in holo form, after the glutamine substrate was removed from the structure, and presented back as a docking ligand using the SwissDock web server. From left to right: (i) the glutamine is docked onto the correct binding pocket within the closed conformation of GlnBP, (ii) amino acid side chains are wrapping the docked glutamine from all directions, (iii) and indeed the protein surface covers the docked glutamine, and (iv) the residues covering the docked glutamine seem to carry a net negative charge.

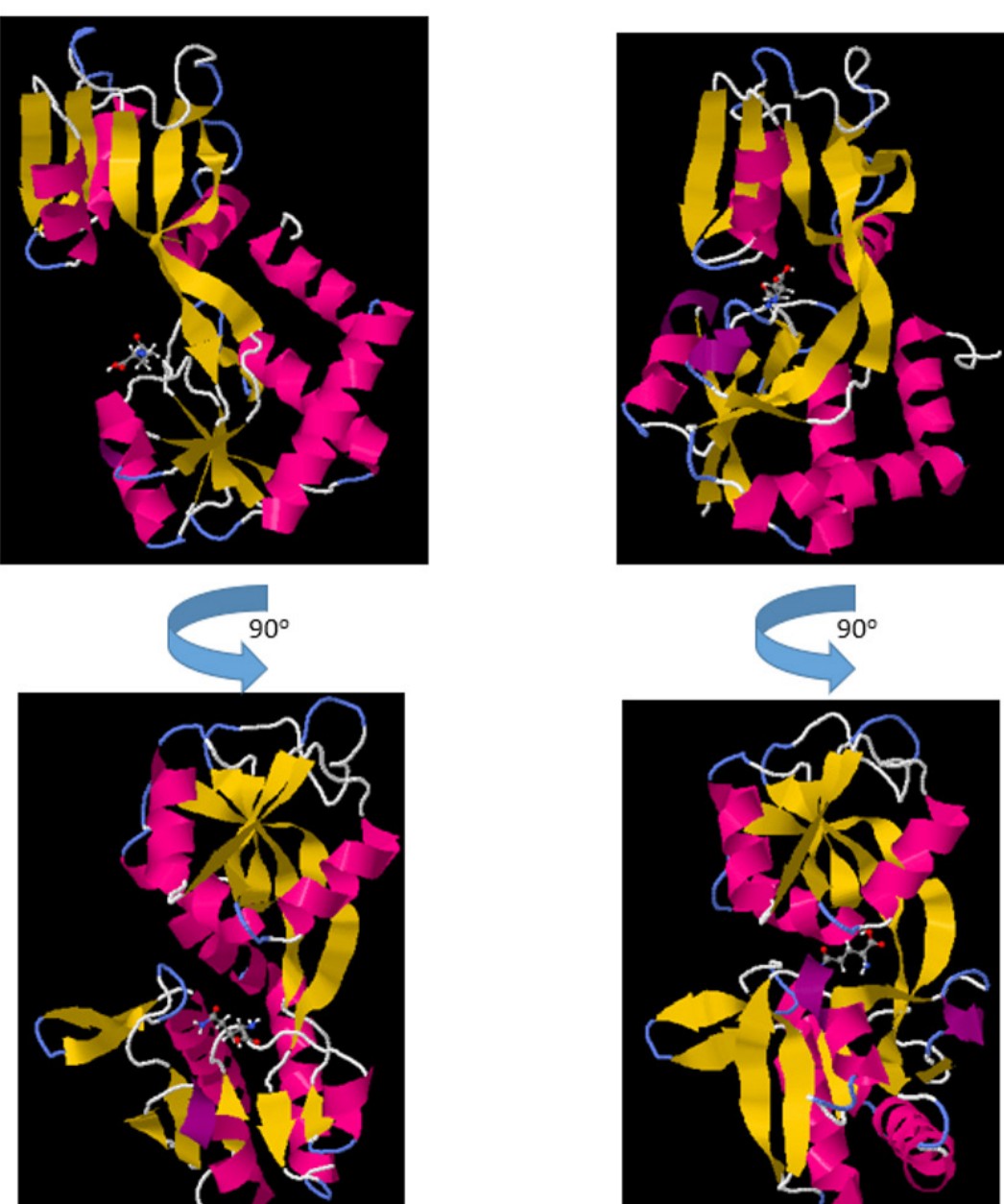

**Appendix 3—figure 2.** Optimized docking of glutamine to GlnBP in its open and closed conformations. Using the SwissDock web server, the molecule glutamine was docked onto the crystal structures of GlnBP in its open (pdb:1GGG) and closed (pdb:1WDN; with the glutamine substrate taken away) conformations, and the optimized docking sites as well as the calculated dissociation constant are shown (dissociation constant is calculated out of the binding energies reported in the docking results). The preferred docking of glutamine is the same site within GlnBP. The difference is that while in the open conformation glutamine binds to one domain with the other as a distant domain, in the closed conformation the other domain closes on top of the docked glutamine. Following the calculated binding energies from the optimized docking results, while the dissociation constant of glutamine to GlnBP is 20 µM in the open conformation, in the closed conformation it is 230 nM.

