## [Editor Report · eLife Assessment]

This **important** study combines a comprehensive range of biophysical, kinetic, and thermodynamic techniques, together with high-quality experimental and computational analysis, to carry out a series of well-designed experiments to explore whether glutamine-binding protein binds glutamine via an induced fit or a conformational selection process. The evidence supporting the major conclusion of the work is **compelling**. The work will be of broad interest to biochemists and biophysicists.

---

## [Referee Report · Reviewer #1 (Public review)]

Here the authors discuss mechanisms of ligand binding and conformational changes in GlnBP (a small E Coli periplasmic binding protein, which binds and carries L-glutamine to the inner membrane ATP-binding cassette (ABC) transporter). The authors have distinguished records in this area and have published seminal works. They include experimentalists and computational scientists. Accordingly, they provide a comprehensive, high quality, experimental and computational work.

They observe that apo- and holo- GlnBP do not generate detectable exchange between open and (semi-) closed conformations on timescales between 100 ns and 10 ms. Especially, the ligand binding and conformational changes in GlnBP that they observe are highly correlated. Their analysis of the results indicates a dominant induced-fit mechanism, where the ligand binds GlnBP prior to conformational rearrangements. They then suggest that an approach resembling the one they undertook can be applied to other protein systems where the coupling mechanism of conformational changes and ligand binding.

They argue that the intuitive model where ligand binding triggers a functionally relevant conformational change was challenged by structural experiments and MD simulations revealing the existence of unliganded closed or semi-closed states and their dynamic exchange with open unbound conformations, discuss alternative mechanisms that were proposed, their merits and difficulties, concluding that the findings were controversial, which, they suggest is due to insufficient availability of experimental evidence to distinguish them. As to further specific conclusions they draw from their results, they determine that a conformational selection mechanism is incompatible with their results, but induced fit is. They thus propose induced fit as the dominant pathway for GlnBP, further supported by the notion that the open conformation is much more likely to bind substrate than the closed one based on steric arguments.

The paper here, which clearly embodies massive careful and high-quality work, is extensive, making use of a range of experimental approaches, including isothermal titration calorimetry, single-molecule Förster resonance energy transfer, and surface-plasmon resonance spectroscopy. The problem the authors undertake is of fundamental importance.

---

## [Referee Report · Reviewer #2 (Public review)]

The authors provide convincing data from a whole set of different binding kinetic and thermodynamic experiments to explore whether glutamine binding protein binds glutamine via an induced fit or a conformational selection process.

Weaknesses:

The single-molecule TIRF-smFRET data appear to include spots that may represent more than one molecule, which raises the general issue of how rigorously traces were selected for single photobleaching events.

---

## [Author Response]

The following is the authors’ response to the original reviews.

**Reviewer #1 (Public Review):**
Here the authors discuss mechanisms of ligand binding and conformational changes in GlnBP (a small E Coli periplasmic binding protein, which binds and carries L-glutamine to the inner membrane ATP-binding cassette (ABC) transporter). The authors have distinguished records in this area and have published seminal works. They include experimentalists and computational scientists. Accordingly, they provide comprehensive, high-quality, experimental and computational work. They observe that apo- and holo- GlnBP does not generate detectable exchange between open and (semi-) closed conformations on timescales between 100 ns and 10 ms. Especially, the ligand binding and conformational changes in GlnBP that they observe are highly correlated. Their analysis of the results indicates a dominant induced-fit mechanism, where the ligand binds GlnBP prior to conformational rearrangements. They then suggest that an approach resembling the one they undertook can be applied to other protein systems where the coupling mechanism of conformational changes and ligand binding. They argue that the intuitive model where ligand binding triggers a functionally relevant conformational change was challenged by structural experiments and MD simulations revealing the existence of unliganded closed or semi-closed states and their dynamic exchange with open unbound conformations, discuss alternative mechanisms that were proposed, their merits and difficulties, concluding that the findings were controversial, which, they suggest is due to insufficient availability of experimental evidence to distinguish them. As to further specific conclusions they draw from their results, they determine that a conformational selection mechanism is incompatible with their results, but induced fit is. They thus propose induced fit as the dominant pathway for GlnBP, further supported by the notion that the open conformation is much more likely to bind substrate than the closed one based on steric arguments. Considering the landscape of substrate-free states, in my view, the closed state is likely to be the most stable and, thus most highly populated. As the authors note and I agree that state can be sterically infeasible for a deep-pocketed substrate. As indeed they also underscore, there is likely to be a range of open states. If the populations of certain states are extremely low, they may not be detected by the experimental (or computational) methods. The free energy landscape of the protein can populate all possible states, with the populations determined by their relative energies. In principle, the protein can visit all states. Whether a particular state is observed depends on the time the protein spends in that state. The frequencies, or propensities, of the visits can determine the protein function. As to a specific order of events, in my view, there isn't any. It is a matter of probabilities which depend on the populations (energies) of the states. The open conformation that is likely to bind is the most favorable, permitting substrate access, followed by minor, induced fit conformational changes. However, a key factor is the ligand concentration. Ligand binding requires overcoming barriers to sustain the equilibrium of the unliganded ensemble, thus time. If the population of the state is low, and ligand concentration is high (often the case in in vitro experiments, and high drug dosage scenarios) binding is likely to take place across a range of available states. This is however a personal interpretation of the data. The paper here, which clearly embodies massive careful, and high-quality work, is extensive, making use of a range of experimental approaches, including isothermal titration calorimetry, single-molecule Förster resonance energy transfer, and surface-plasmon resonance spectroscopy. The problem the authors undertake is of fundamental importance.
**Reviewer #2 (Public Review):**
The manuscript by Han et al and Cordes is a tour-de-force effort to distinguish between induced fit and conformational selection in glutamine binding protein (GlnBP).

We thank the referee for the recognition of the work and effort that has gone into this manuscript.

It is important to say that I don't agree that a decision needs to be made between these two limiting possibilities in the sense that whether a minor population can be observed depends on the experiment and the energy difference between the states. That said, the authors make an important distinction which is that it is not sufficient to observe both states in the ligand-free solution because it is likely that the ligand will not bind to the already closed state. The ligand binds to the open state and the question then is whether the ligand sufficiently changes the energy of the open state to effectively cause it to close. The authors point out that this question requires both a kinetic and a thermodynamic answer. Their "method" combines isothermal titration calorimetry, single-molecule FRET including key results from multi-parameter photon-by-photon hidden Markov modelling (mpH2MM), and SPR. The authors present this "method" of combination of experiments as an approach to definitively differentiate between induced fit and conformational selection. I applaud the rigor with which they perform all of the experiments and agree that others who want to understand the exact mechanism of protein conformational changes connected to ligand binding need to do such a multitude of different experiments to fully characterize the process. However, the situation of GlnBP is somewhat unique in the high affinity of the Gln (slow offrate) as compared to many small molecule binding situations such as enzyme-substrate complexes. It is therefore not surprising that the kinetics result in an induced fit situation.

For us these comments are an essential part of the conceptual aspects of our work and the resulting research. From a descriptive viewpoint, it is essential for us (and we tried to further highlight and stress this in the updated version of our paper) that IF and CS are two kinetic mechanisms of ligand binding. They imply – if active in a biomolecular system – a temporal order and timescale separation of ligand binding and conformational changes. Since we found many conflicting results for the binding mechanism of GlnBP, but also other SPBs, we decided to assess the situation in GlnBP.

In the case of the E-S complexes I am familiar with, the dissociation is much more rapid because the substrate binding affinity is in the micromolar range and therefore the re-equilibration of the apo state is much faster. In this case, the rate of closing and opening doesn't change much whether ligand is present or not. Here, of course, once the ligand is bound the re-equilibration is slow. Therefore, I am not sure if the conclusions based on this single protein are transferrable to most other protein-small molecule systems.

We do not argue that our results and interpretations are valid for most other protein-ligand systems may those be enzymes or simple ligand binders. Yet, based on the conservation of ABC-related SBPs and the fact that quite a few of them show sub-µM Kds, we render it likely to find many analogous situations as for GlnBP also based on our previous results e.g., from de Boer et al., eLife (2019).

I am also not sure if they are transferrable to protein-protein systems where both molecules the ligand and the receptor are expected to have multiscale dynamics that change upon binding.

As we argue above the two mechanisms IF/CS imply a clear temporal order and separation of timescales for ligand binding and conformational changes. These mechanisms are simple and extreme cases that we tested before more complex kinetic schemes are inferred for the description of ligand binding and conformational changes (which might not be necessary).

Strengths:The authors provide beautiful ITC data and smFRET data to explore the conformational changes that occur upon Gln binding. Figure 3D and Figure 4 (mpH2MM data) provide the really critical data. The multi-parameter photon-by-photon hidden Markov modelling (mpH2MM) data. In the presence of glutamine concentrations near the Kd, two FRET-active sub-populations are identified that appear to interconvert on timescales slower than 10 ms. They then do a whole bunch of control experiments to look for faster dynamics (Figure 5). They also do TIRF smFRET to try to compare their results to those of previous publications. Here, they find several artifacts are occurring including inactivation of ~50% of the proteins. They also perform SPR experiments to measure the association rate of Gln and obtain expectedly rapid association rates on the order of 10^^^8 M-1s-1.

Thank you.

Weaknesses:Looking at the traces presented in the supplementary figures, one can see that several of the traces have more than one molecule present. The authors should make sure that they use only traces with a single photobleaching event for each fluorophore. One can see steps in some of the green traces that indicate two green fluorophors (likely from 2 different molecules) in the traces. This is one of the frequent problems with TIRF smFRET with proteins, that only some of the spots represent single molecules and the rest need to be filtered out of the analysis.

We have inspected all TIRF data provided with the manuscript and assume that the referee refers to data shown in current Appendix Figure 4/5. We agree that those traces in which no photo bleaching occurs could potentially be questioned, yet they would not change our interpretations and thus decided to leave the figure as is.

The NMR experiments that the authors cite are not in disagreement with the work presented here. NMR is capable of detecting "invisible states" that occur in 1-5% of the population. SmFRET is not capable of detecting these very minor states. I am quite sure that if NMR spectroscopists could add very high concentrations of Gln they would also see a conversion to the closed population.

We agree with the referee that NMR is capable of detecting invisible states that occur in 1-5% of the population (see e.g., the paper cited in our manuscript by Tang, C et al., Open-to-closed transition in apo maltose-binding protein observed by paramagnetic NMR. Nature 2007, 449, 1078). Yet, we see a strong disagreement between our work and papers on GlnBP, where a combination of NMR, FRET and MD was used (Feng, Y. et al., Conformational Dynamics of apo‐GlnBP Revealed by Experimental and Computational Analysis. Angewandte Chemie 2016, 55, 13990; Zhang, L. et al., Ligand-bound glutamine binding protein assumes multiple metastable binding sites with different binding affinities. Communications biology 2020, 3, 1). These inconsistencies were also noted by others in the field (Kooshapur, H. et al., NMR Analysis of Apo Glutamine‐Binding Protein Exposes Challenges in the Study of Interdomain Dynamics. Angewandte Chemie 2019, 58, 16899) and we reemphasize that this latest NMR publication comes to similar conclusions as we present in our manuscript.

**Reviewer #1 (Recommendations For The Authors):**
The paper embodies massive careful and high-quality work, and is extensive, making use of a range of experimental approaches, including isothermal titration calorimetry, single-molecule Förster resonance energy transfer, and surface-plasmon resonance spectroscopy. Considering this extensiveness, I do not see what more the authors can do.

We very much appreciate the assessment and positive comments of the referee, but still tried to incorporate simulation data to support our interpretations.

**Reviewer #2 (Recommendations For The Authors):**
(1) Looking at the traces presented in the supplementary figures, one can see that several of the traces have more than one molecule present. The authors should make sure that they use only traces with a single photobleaching event for each fluorophore. One can see steps in some of the green traces that indicate two green fluorophors (likely from 2 different molecules) in the traces. This is one of the frequent problems with TIRF smFRET with proteins, that only some of the spots represent single molecules and the rest need to be filtered out of the analysis.

See response above for iteration of TIRF data selection and analysis.

(2) The NMR experiments that the authors cite are not in disagreement with the work presented here. NMR is capable of detecting "invisible states" that occur in 1-5% of the population. SmFRET is not capable of detecting these very minor states. I am quite sure that if NMR spectroscopists could add very high concentrations of Gln they would also see a conversion to the closed population.

See response above.

Minor point:(1) It is difficult to see what is going on between apo and holo in Figure 1B. Could the authors make Figure 1a, 1b apo, and 1b holo in the same orientation (by aligning D2 or D1 to each other in all figures) so one can see which helices are in the same place and which have moved?

We respectfully disagree and decided to keep this figure as it is